# Restoration of lysosomal acidification rescues autophagy and metabolic dysfunction in non-alcoholic fatty liver disease

Jialiu Zeng ®[1,2] ✉, Rebeca Acin-Perez[3,10], Essam A. Assali ®[3,10], Andrew Martin[1], Alexandra J. Brownstein[3], Anton Petcherski[3], Lucía Fernández-del-Rio ®[3], Ruiqing Xiao[4,5], Chih Hung Lo ®[2], Michaël Shum[3,6], Marc Liesa ®[3,6,7,8], Xue Han ®[1], Orian S. Shirihai ®[3,6,9] ✉ & Mark W. Grinstaff ®[1,4,9] ✉

Non-alcoholic fatty liver disease (NAFLD) is the most common liver disease in the world. High levels of free fatty acids in the liver impair hepatic lysosomal acidification and reduce autophagic flux. We investigate whether restoration of lysosomal function in NAFLD recovers autophagic flux, mitochondrial function, and insulin sensitivity. Here, we report the synthesis of novel bio-degradable acid-activated acidifying nanoparticles (acNPs) as a lysosome targeting treatment to restore lysosomal acidity and autophagy. The acNPs, composed of fluorinated polyesters, remain inactive at plasma pH, and only become activated in lysosomes after endocytosis. Specifically, they degrade at pH of ~6 characteristic of dysfunctional lysosomes, to further acidify and enhance the function of lysosomes. In established in vivo high fat diet mouse models of NAFLD, re-acidification of lysosomes via acNP treatment restores autophagy and mitochondria function to lean, healthy levels. This restoration, concurrent with reversal of fasting hyperglycemia and hepatic steatosis, indicates the potential use of acNPs as a first-in-kind therapeutic for NAFLD.

Non-alcoholic fatty liver disease (NAFLD) affects 20 to 30% of the world's population[1]. NAFLD is characterized by the accumulation of lipid in the liver, and its prevalence directly correlates with obesity and insulin resistance. Patients with untreated NAFLD subsequently develop non-alcoholic steatohepatitis with some fibrosis, which can progress to liver cirrhosis[2]. Continued decline in liver function leads to hepato-carcinoma or liver failure, where the only treatment option is a liver transplant, which has a low likelihood of success[2]. Clinically today, no approved pharmacological agents target the liver directly to counteract NAFLD[3,4]. Instead, current treatments act by increasing systemic insulin sensitivity such as thiazolidinediones, rosiglitazone, metformin[3], and/or approaches such as decreasing food intake, for instance GLP-1 receptor agonism (exenatide and liraglutide)[3,5–7]. However, these treatments possess unwanted off-target effects and involve perennial dosing.

[1]Department of Biomedical Engineering, Boston University, Boston, MA 02215, USA. [2]Lee Kong Chian School of Medicine, Nanyang Technological University, Singapore, 308232 Singapore, Singapore. [3]Division of Endocrinology, Department of Medicine, David Geffen School of Medicine, University of California, Los Angeles, Los Angeles, CA 90045, USA. [4]Department of Chemistry, Boston University, Boston, MA 02215, USA. [5]Shenzhen Middle School, Shenzhen, Guangdong 518001, China. [6]Department of Molecular and Medical Pharmacology, University of California, Los Angeles, Los Angeles, CA 90095, USA. [7]Molecular Biology Institute at University of California, Los Angeles, Los Angeles, CA 90095, USA. [8]Institut de Biologia Molecular de Barcelona, IMBB, CSIC, Barcelona, Catalonia 08028, Spain. [9]Department of Medicine, Boston University Chobanian & Avedisian School of Medicine, Boston, MA 02118, USA. [10]These authors contributed equally: Rebeca Acin-Perez, Essam A. Assali. ✉e-mail: jialiu.zeng@ntu.edu.sg; OShirihai@mednet.ucla.edu; mgrin@bu.edu

With regards to obesity and type 2 diabetes-associated NAFLD, insulin resistance in adipose tissue hinders insulin-mediated suppression of lipolysis and, thus, increases the levels of serum free fatty acids (FFA)[8]. FFAs are metabolized by the liver and esterified into neutral triglycerides with accumulation of lipid droplets; however, an excess of saturated FFAs overwhelms the capacity of the liver to esterify FFA affording hepatocyte toxicity (i.e., lipotoxicity)[8,9]. Under lipotoxicity, FFA inhibition of lysosomal function blocks autophagic flux, which causes the accumulation of damaged mitochondria as well as of lipids digested by lipophagy, thereby exacerbating the decline in mitochondria and cellular functions, which leads to progressive disease[8,10–15,16–18]. Autophagy is an essential cellular maintenance mechanism[17], and two conditions are necessary for the maintenance of autophagy: 1) autophagosome recruitment and engulfment of the cellular contents to be degraded; and, 2) autophagosome fusion with a sufficiently acidic lysosome whose pH is tightly regulated by the vacuolar ATPase (V-ATPase)[19–22]. Consequently, restoring lysosomal acidification and autophagic flux has the potential to reverse NAFLD, and represents a novel treatment strategy (Fig. 1). Specifically, we hypothesize that by lowering the pH of impaired lysosomes via delivery of nanoparticles, autophagic flux and cellular function will be restored and in vivo function will be rescued, as measured by a reduction in liver triglyceride levels and an improved glucose and insulin tolerance response. Nanoparticles, which are particulate materials typically around 50–200 nm in diameter, are increasingly used to deliver therapeutic agents or as a tool to modulate cellular processes[23–27].

Herein, we report a new type of responsive polymeric nanoparticle, acid-activated acidic nanoparticle (acNP), that successfully restores lysosomal acidification, and reverses the manifestations of obesity induced NAFLD in mice.

## Results

### AcNPs efficiently endocytose and localize to lysosomes of HepG2 cells

A key design criterion of the acid-activated acidic nanoparticles (acNPs) is the selective acidification of the local lysosomal interior environment upon degradation. Thus, we chose to use NPs composed of a polyester, however polyesters, in general, do not readily hydrolyze in aqueous environment at pH 7–7.4. Yet, in the presence of a slightly acidic environment (pH 6), the rate of hydrolysis and degradation increases, resulting in the release of protons. The challenge lies in the additional lowering of the poorly functioning lysosomal pH of 6 upon hydrolysis and NP degradation. Previously synthesized biodegradable polyesters composed of acids such as glycolic acid or lactic acid, which have $pK_a$ values of 3.83 and 3.86, respectively, only slightly lower the pH[28]. However, by incorporating a more potent acid, which possesses a $pK_a$ two to three orders of magnitude lower than 4 into the polymer, a greater acidic response will be generated. We selected a fluorinated di-acid, tetra-fluorosuccinic acid (TFSA) as it has a $pK_a$ of -1.6[29]. Fluorinated polyesters exhibit low toxicity and high in vitro and in vivo biocompatibility, and have been extensively used in previous medical applications[30,31]. To demonstrate the acidification and pH lowering capabilities of the acNPs, we synthesized two different polyesters from tetrafluorosuccinic acid (TFSA), succinic acid (SA), and ethylene glycol for NP fabrication. We varied the ratio between TFSA and SA— one composed entirely of SA (PESU) and the other composed of 5–25 mol% TFSA and 75 mol% SA (PEFSU) (Fig. 2A). Our attempts to prepare polymers with greater TFSA mol% were unsuccessful. The polymers were synthesized in greater than 90% yield via a dehydration polymerization reaction, and subsequently characterized via

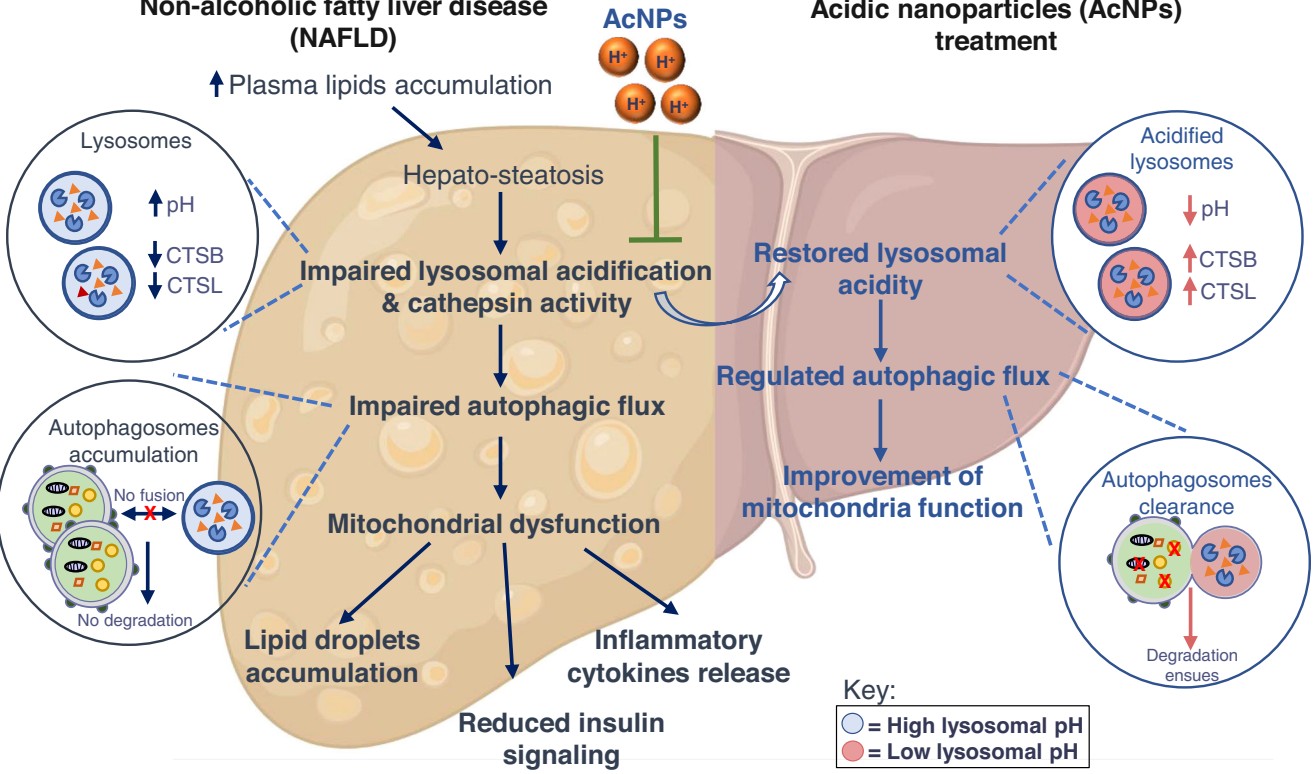

**Fig. 1 | The role of defective lysosomal function in NAFLD.** In an NAFLD liver, diet-induced obesity leads to increased plasma lipids accumulation, contributing to hepatic steatosis and lipotoxicity. Lipotoxicity affords impaired lysosomal acidification, lysosomal dysfunction, autophagic degradation dysfunction, and subsequent leads to impaired mitochondria function and lipid droplets accumulation.

We hypothesize that acid nanoparticle (acNP) treatment inhibits impaired lysosomal acidification through restoration of lysosomal acidity, thereby restoring autophagic and mitochondria function, with a resultant decrease in lipid droplets accumulation and steatosis in NAFLD. This figure is created with BioRender.com<http://BioRender.com>.

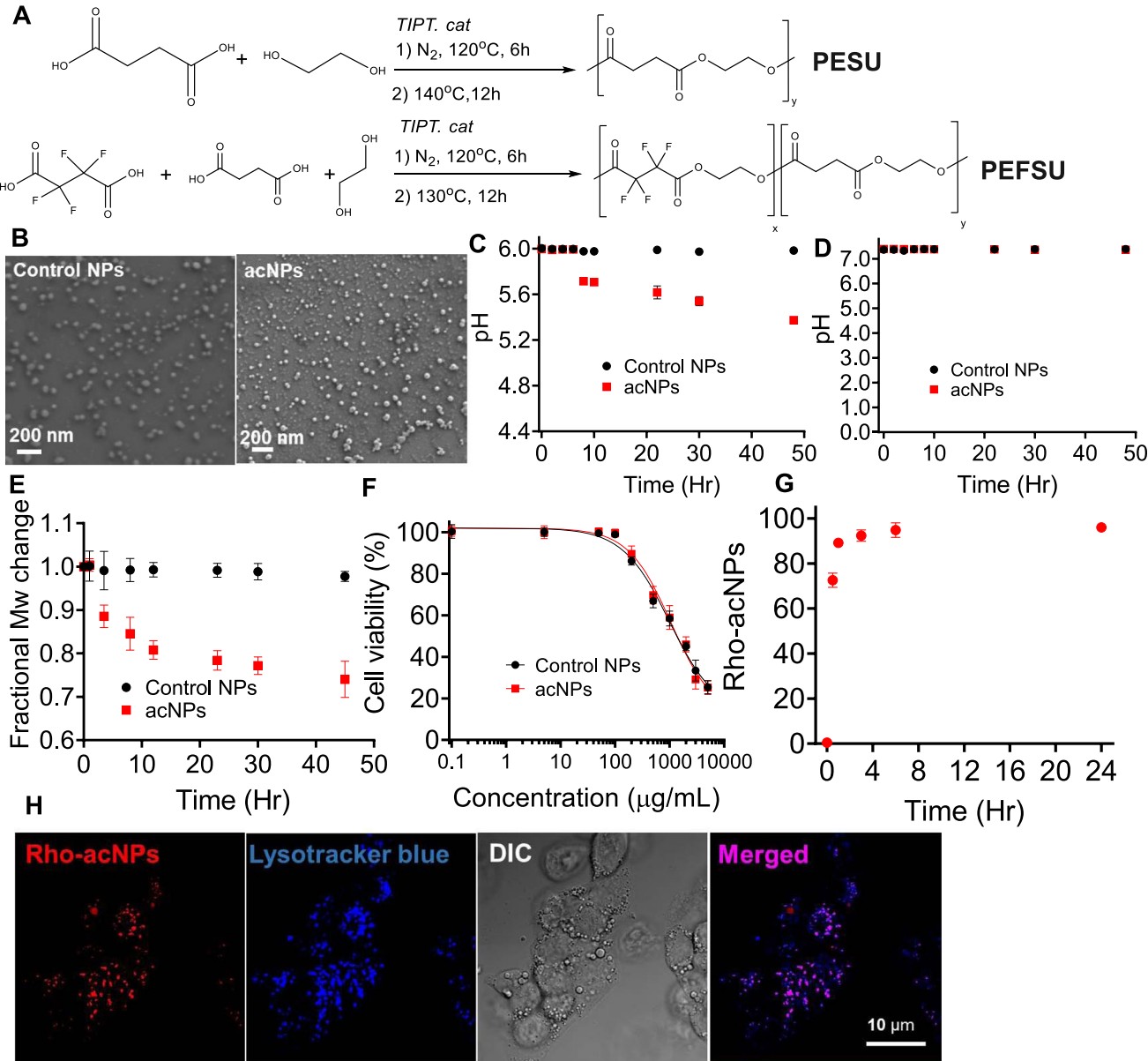

**Fig. 2 | Synthesis and characterization of acNPs. A** Synthesis route of PESU and PEFSU polymer using polycondensation. **B** Scanning electron micrograph images of the control NPs and acNPs show that they are of spherical morphology and size around 100 nm ($N = 3$ independent experiments). Bar, 200 nm. **C** pH changes of acNPs in 20 mM pH 6.0 and **D** pH 7.4 PBS buffers over a period of 48 h ($N = 3$ independent experiments). acNPs significantly acidified buffer within the first 24 h. **E** Fractional molecular weight changes of control NPs and acNPs in 20 mM pH 6.0 PBS buffer ($N = 3$ independent experiments). **F** Dependency of acNP toxicity on dose. Cells were incubated with a range of acNP concentrations for 24 h. acNPs up

to 1000 μg/mL did not induce significant cell death. To avoid cytotoxicity, a treatment dose of 100 μg/mL was chosen for further studies ($N = 3$ independent experiments). **G** Quantification of rhodamine-labeled acNP (Rho-acNP) uptake in HepG2 cells by flow cytometry. Cellular uptake of Rho-acNPs occurs within 4 h, with complete uptake after 24 h of incubation ($N = 3$ independent experiments). **H** Representative confocal microscopy images of Rho-acNPs in HepG2 cells. Rho-acNPs (red channel) localize within the lysosomal compartments (LysoTracker; blue channel) of HepG2 cells. Bar, 10 μm ($N = 3$ independent experiments). Data are expressed as means ± SD. Source data are available as a Source Data file.

nuclear magnetic resonance spectroscopy and gel permeation chromatography (Supplementary Figs. 1A–C and 2A).

We hypothesized that NPs composed of 25 % PEFSU (acNPs) will significantly lower the surrounding pH as they degrade, while the PESU NPs will cause no significant decrease in pH, and, thus, serve as a control NP for our studies. We prepared monodisperse NPs of these polymers using a nanoprecipitation technique and sodium dodecyl sulfate as the surfactant[32]. The NPs exhibit average diameters around 100 nm by scanning electron microscopy (SEM) (Fig. 2B) and dynamic light scattering, as well as possess a zeta potential (ζ) between −25 to −30 mV (Supplementary Fig. 2B). In a 20 mM pH 6.0 phosphate buffered saline (PBS) solution to mimic the lysosomal environment[33], the

PEFSU acNPs significantly acidify the solution within the first 4 h and continue to acidify the solution for over the next 40+ h (Fig. 2C). The final pH of the solution is 5.3. In contrast, the addition of control NP (i.e., PESU) to this buffer does not result in a lowering of the pH. Similarly, acNPs composed of lesser amounts of TFSA exhibit reduced pH acidification capabilities. For comparison, addition of PLGA NPs to the same pH 6.0 buffer affords only a modest reduction in pH to 5.9 and the change occurs slowly over 48 h (Supplementary Fig. 2C, D)[28, 34,35]. When the acNPs are exposed to a pH 7.4 PBS solution, the pH minimally decreases, indicating constitutive activation at pH 6 (Fig. 2D). In addition, we investigated the molecular weight changes over time of the control and PEFSU acNPs in 20 mM pH 6.0 PBS buffer.

The acNPs degrade rapidly over the first 24 h, followed by a much slower rate of degradation within the next 24 h (Fig. 2E). SEM micrographs of the acNPs as well as quantification of nanoparticles sizes show that there is an increase in size/swelling within the first 24 h, indicative that the PEFSU polymer is undergoing bulk degradation (Supplementary Fig. 2E, F). We also studied the correlation between polymer degradation and pH release through quantifying the amount of TFSA or SA released (Supplementary Fig. 2G). The calculations reveal that the amount of SA released by PESU is insufficient to result in pH changes, while addition of 25 mol% TFSA in PEFSU significantly lowers the pH (Supplementary note). To determine the optimal acNP concentration to treat HepG2 cells without inducing cell death, we performed dose response cell cytotoxicity assays across a concentration range from 10 µg/mL to 5000 µg/mL (24 h exposure). PEFSU acNPs do not result in significant cell death up to a concentration of 100 µg/mL (Fig. 2F). We selected an optimal dose of 100 µg/mL for further assays to avoid cytotoxicity. Rhodamine-labeled PEFSU acNPs (Rho-acNPs) are rapidly taken up in HepG2 cells with 80% of the HepG2 cells possessing Rho-acNPs within 4 h as revealed by flow cytometry (Fig. 2G). Next, we performed confocal microscopy to confirm uptake of the acNPs into the lysosome compartment after incubating the HepG2 cells with LysoTracker blue dye and Rho-acNPs for 24 h. As shown in Fig. 2H, co-localization is clear. These results are consistent with negatively charged nanoparticles of ~100 nm in diameter being rapidly uptaken and localizing within the endosome-lysosome system[36,37].

## AcNPs re-acidify lysosomes and restore autophagic flux in HepG2 cells under lipotoxicity

Having demonstrated acNP function and cellular uptake into lysosomes, the essential subsequent question is whether acNPs restore lysosomal acidity in lipotoxic cells exposed to palmitate—a long-chain fatty acid. Thus, we exposed HepG2 cells to either BSA control, 400 µM palmitate (complexed to BSA at 4:1 ratio)[38], or 400 µM palmitate with 100 µg/mL acNPs for 16 h (Fig. 3A). Following incubation, we assessed lysosomal acidity and size by confocal imaging with Lysosensor dye (Fig. 3B). Palmitate exposure to the HepG2 cells significantly increases lysosomal pH and size compared to BSA-treated cells by a magnitude of 0.6 pH units and 8.7 units[2], respectively. Addition of acNPs to the BSA control does not alter lysosomal pH or size (Supplementary Fig. 3A, B). Treatment of HepG2 cells with acNPs decreases the lysosomal pH by 0.4 units (Fig. 3C) and reduces the average lysosomal size (Fig. 3C). In addition, the number of lysosomes across different treatment conditions does not change significantly (Supplementary Fig. 3C). These changes are consistent with increased lysosomal turnover by elevated autophagic flux[39]. In contrast, addition of control NPs does not change lysosomal pH, due to its inability to release as many protons. Furthermore, acNP treatment significantly restores cathepsin L activity in cells treated with palmitate[16,39], supporting that re-acidification of lysosomes recovers their proteolytic function (Fig. 3D & Supplementary Fig. 3D).

Next, we tested whether the restoration of lysosomal acidity induced by acNPs improves the autophagic flux. To measure autophagic flux, we quantified the levels of autophagic substrates, such as microtubule-associated protein 1A/1B light chain 3 (LC3-II) and p62, in HepG2 cells in the presence or absence of Bafilomycin A1 (Baf) by western blotting[40, 41]. No change in the autophagic flux occurs across BSA, BSA and control NPs, and BSA and acNPs treated conditions (Fig. 3E–G). Addition of palmitate reduces autophagic flux and treatment with acNPs rescues this effect (Fig. 3E–G). This result confirms that the restoration of lysosomal acidification by acNPs reverses the blockage in autophagic flux induced by palmitate treatment. As autophagy is implicated in the removal of lipids by lipophagy in liver in vivo, we tested whether acNPs decrease the lipid load in hepatocytes in vitro and in vivo. Exposure of HepG2 cells to palmitate increases the

number of lipid droplets significantly to 20–25 per cell (Fig. 3H) compared to 10 under BSA control. Treatment of HepG2 cells with acNPs reduces the accumulation of lipid droplets to 15 droplets per cell (Fig. 3I) and decreases the total triglyceride levels (Supplementary Fig. 3E). Additionally, palmitate treatment results in ~20% cell death, while treatment of acNPs at 100 µg/mL is protective (Supplementary Fig. 3F, G). We next determined if acNP treatment modulates HepG2 insulin sensitivity due to a decrease in FFA-mediated impairments in insulin signaling via immunoblotting. The results are reported as insulin-stimulated values compared with unstimulated or basal values. Palmitate treatment prevents insulin from increasing the phosphorylation (p) levels of insulin receptor β (IR; tyrosine 1162/1163) and protein kinase B (Akt; serine 473), indicators of insulin signaling activation in HepG2 cells (Supplementary Fig. 3H–J). The lack of liver glycogen has been shown to lead to increased fat accumulation and the development of liver insulin resistance[42,43]. Glycogen synthesis is induced through Akt inhibition of Glycogen synthase kinase-3 (GSK3). Under palmitate treatment, reduced Akt activation leads to decreased phosphorylation levels of GSK3β serine 9, resulting in reduced glycogen synthesis (Supplementary Fig. 3H, K). acNP treatment rescues this deficiency through increasing insulin-stimulated p-IR (Tyr 1162/1163), p-Akt (Ser 473), and p-GSK3β (Ser 9) levels compared to palmitate treated group, indicating an improvement in insulin sensitivity in HepG2 cells (Supplementary Fig. 3H, K). Given that decreased autophagy due to inhibition of lysosomal function leads to the accumulation of dysfunctional mitochondria, we assessed the mitochondria respiration function of HepG2 cells under BSA, palmitate, or palmitate with acNP treatments. Palmitate treatment reduces the mitochondria oxygen consumption rate (OCR) compared to BSA control, where addition of acNPs restore mitochondria activity (Fig. 3J).

## AcNPs improve NAFLD in high-fat diet-fed mice

Having shown promising therapeutic efficacy of acNPs in lipotoxic HepG2 cells, we investigated acNP efficacy in vivo using a high-fat diet (HFD) induced NAFLD mouse model, where C57BL/6 mice are fed with a HFD for 16 weeks. First, we determined the in vivo biodistribution and localization of acNPs in mice. We intravenously (i.v.) administered Rho-acNPs of around 100 nm in diameter and harvested the tissues 24 h after injection to quantify for rhodamine fluorescence in tissue lysates, expressed as fold change to vehicle injection. acNPs primarily accumulate in the liver and kidney tissues, which are both known sites of nanoparticles accumulation and clearance (Supplementary Fig. 4A, B)[44,45]. AcNPs do not accumulate in other major organs including lungs, heart, brain, and white adipose tissue (WAT). Next, we determined how much of liver acNPs uptake was mediated by Kupffer cells, as it is known that NPs are taken up by Kupffer cells after tail vein injection[46,47]. 20% of the hepatic Rho-acNPs localize to Kupffer cells, as measured by the co-localization of Rhodamine-red with anti-CLEC4F immunofluorescent signal in liver tissue sections[48] (Supplementary Fig. 4C). LAMP-1 lysosomal staining also shows that Rho-acNPs localize in the lysosomes detected in liver sections (Supplementary Fig. 4D).

To determine safety and optimal dosage of acNPs, we first i.v. injected 100 mg/kg/day (low dose, LD) or 300 mg/kg/day (high dose, HD) either as a single or three administrations over six days (injection regimen shown in Fig. 4A). Treatment with acNPs did not result in significant weight loss (Supplementary Fig. 4E), reduction in food intake (Supplementary Fig. 4F), alterations of blood chemistry (Supplementary Tables 1 and 2), or increases in serum alanine aminotransferase (ALT) and bilirubin (BIL) levels (Fig. 4B–G)—clinical indicators of liver damage. In fact, multiple injections of HD acNPs decrease serum ALT and BIL levels compared to HFD treated mice (Fig. 4B–D). Multiple injections of either LD or HD acNPs significantly decrease serum triglyceride levels (Fig. 4G)—while a single injection of acNPs does not afford the same result (Fig. 4D). These findings indicate that more than one injection of acNPs is required for efficacy and that

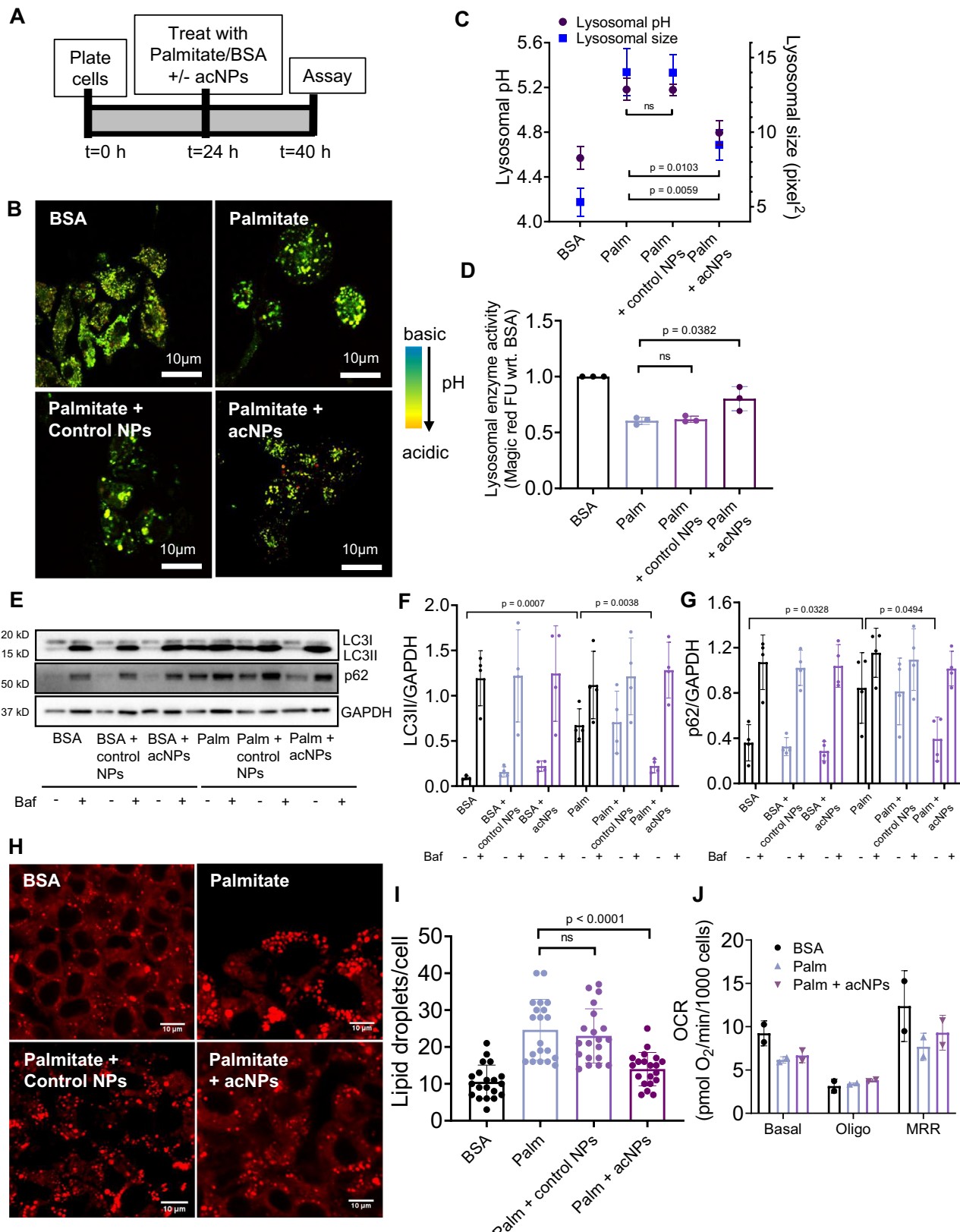

acNPs protect against liver steatosis in vivo. Replication of this study using a different mouse sub-strain C57BL/6N with 17 weeks of HFD[49] at a different facility yielded similar outcomes (Supplementary Fig. 5A–H).

Next, we determined if acNP treatment changes systemic glucose control, by performing glucose tolerance (ipGTT) and insulin tolerance tests (ipITT). As expected, HFD increases fasting insulin and

c-peptide levels (Fig. 5A, B), indicative of insulin resistance. In the LFD group, acNPs exhibit no effects on the GTT and ITT profiles, as expected with an absence of autophagy deficits in LFD-fed mice (Fig. 5C, D). Treatment of HFD mice with LD or HD acNPs significantly improves glucose clearance after 30 min in a dose dependent response, which is compatible with an improvement in hepatic

**Fig. 3 | Treatment of HepG2 with acNPs reverses the defects in lysosomal acidity, cathepsin L activity and autophagic flux induced by lipotoxicity.**
**A** Schematic of experimental protocol for cell treatment for 16 h before assaying for lysosomal acidity, autophagy, or cellular function. The indicated conditions are control BSA, 400 μM palmitate complexed to BSA, or 400 μM palmitate with control NPs and acNPs treatment. The 16 h timepoint is chosen because it shows the highest amount of autophagic flux inhibition in HepG2 cells under palmitate.
**B** Representative confocal microscopy images of HepG2 cells treated with the indicated conditions and stained with 75 nM pH-sensitive Lysosensor dye to assess lysosome acidity. Bar, 10 μm. **C** Mean lysosomal pH and lysosomal area per cell for cells exposed to the indicated conditions were analyzed by MetaMorph® show significant restoration of lysosomal pH compared with palmitate cells ($N = 3$ independent experiments with $n = 20$ cells analyzed per condition). **D** Assessment of lysosomal cathepsin L activity by Magic red fluorescent substrate assay in HepG2

cells exposed to all the conditions showed significant restoration of lysosomal enzyme activity with acNPs treatment but not with control NPs treatment ($N = 3$ independent experiments). **E–G** Representative western blot exposure images showing protein expression data for control NPs and acNPs treated to HepG2 cells, before and after addition of Bafilomycin A1 (Baf) ($N = 4$ independent experiments). **H** Representative confocal images of HepG2 cells stained with Nile Red dye for 15 min and imaged with fluorescence microscopy. Nile red dye accumulated quickly in the lipid vesicles. Bar, 10 μm. **I** Quantification of lipid vesicles number indicated significant reduction in lipid droplets density after acNPs treatment in HepG2 cells exposed to palmitate. Control NPs addition did not reduce lipid droplets ($n = 20$ cells analyzed per condition). **J** Mitochondria oxygen consumption rates in HepG2 cells under BSA, palmitate or palmitate with acNPs ($N = 3$ independent experiments). Two-tailed unpaired $t$-test (**C**, **D**, **F**, **G**, **I**, **J**); data are expressed as means ± SD, n.s. not statistically significant. Source data are available as a Source Data file.

glycogen synthesis (Fig. 5E). To account for differences in baseline fasting glucose levels measured across the treatment groups, we calculated the area under the ipGTT curve, and HD acNPs significantly decrease total glucose excursion compared to HFD control (Fig. 5F), indicative of improved glucose response. For the ipITT experiment, we injected insulin into LD or HD acNP treated animals and recorded the insulin tolerance response over 2 h (Fig. 5G). The ipITT results reveal that in LD and HD treated mice, insulin more effectively decreases glucose levels (Fig. 5G). Thus, the GTT and ITT glycemic profiles are consistent with an improved in hepatic glucose metabolism.

### AcNPs reverse liver steatosis in high-fat diet-fed mice
Anatomically, HD acNP treatment reduces the liver weight (Fig. 6A) and liver weight to body weight ratio in HFD-fed mice (Supplementary Fig. 6A), but not in LFD-fed mice (Fig. 6B & Supplementary Fig. 6B). The liver weights of mice and liver histology, under different treatments for the ipGTT and ipITT experiments (Supplementary Figs. 6C–J and 7A–C), maintain the weight reduction even after one week without additional acNP injections, indicating that the effect is not short-term. To further investigate if this reduction in liver weight is due to reduced lipid accumulation, we quantified liver triglyceride levels in total liver lysates. Multiple injections of LD or HD acNPs reduce triglyceride levels in HFD-fed mice, with HD acNPs exhibiting greater potency than LD acNPs (Fig. 6C). Additionally, there is no significant change in the visceral body fat pads or body weight before and after acNP injections (Supplementary Fig. 7D), indicating that the metabolic benefits of acNPs are not a result of protecting from obesity. We assessed liver steatosis using a fatty liver scoring grid of hematoxylin and eosin stained slides (H & E)[50] (Supplementary Table. 3). In the HFD control group, liver sections show significant areas of enlarged hepatocytes, and lipid droplets (Fig. 6D) indicative of grade 2 (macro) steatosis (> 33–66% area with lipid droplets) (Fig. 6G). Treatment with single injections of LD or HD acNPs does not reverse this effect significantly, as both sections still scored grade 1 steatosis and grade 2 steatosis, respectively (Fig. 6D, G). In contrast, mice receiving multiple injections of LD or HD acNPs show significant reduction in lipid droplet accumulation (Fig. 6E) and a steatosis scored grade 0 (Fig. 6G). Additionally, microvesicular steatosis is no longer present. While the LD acNPs do not induce any toxicity, small amounts of lobular inflammation (<2 foci per 200x field) are observed in HD acNPs, although not deemed as severe by the pathologist. To demonstrate that the effect of acNPs is specific to HFD mice, we administered multiple injections of LD or HD acNPs to LFD mice. H & E stain of mice liver slices shows that LD or HD acNPs do not result in any significant change in steatosis compared to LFD mice receiving a saline injection (Fig. 6F, G).

### acNPs improve mitochondria function and decrease expression of inflammatory markers via restoration of autophagic flux
We next investigated whether autophagic flux rescues livers from high-fat diet-fed mice, as we observed in HepG2 hepatocytes in vitro

(Fig. 3F, G). A significant decrease occurs in both LC3-II and p62 in HFD mice treated with acNPs (Fig. 7A–C), indicating restoration of autophagic flux. In contrast, treatment with acNPs does not change LC3-II and p62 levels in LFD mice, as expected, given that they do not show defects in autophagic flux or lipotoxicity (Fig. 7D–F). These same results were replicated using a different strain (C57BL/6N) housed in a different facility (Supplementary Fig. 8A–C). The restoration of LC3-II and p62 levels is concurrent with increased lysosomal enzyme activity (Supplementary Fig. 8D).

Mitochondria play a key role in the development of NAFLD, with some genetic models inducing a decrease in mitochondrial respiratory capacity showing an exacerbation in HFD-induced NAFLD[51, 52]. Impairment in lysosomal acidification is known to decrease mitochondrial respiration[53,54]. Treatment with either LD or HD acNPs does not significantly change mitochondrial content in mice total liver lysates (Fig. 7G, H), consistent with the absence of a coordinated reduction in mitochondria OXPHOS complex subunits (Supplementary Fig. 8E, F). Next, to test whether lysosomal acidification by acNPs leads to an improvement in mitochondria function, we measured total mitochondrial respiratory function in fresh mice liver homogenates, using a published method[55]. AcNP treatment increases mitochondrial OCR fueled by pyruvate and malate under state 3 (maximal ATP synthesis capacity), under oligomycin (leak) and under FCCP, demonstrating increased respiratory capacity (Fig. 7I). Remarkably, when the fuel provided to mitochondria was succinate and rotenone or palmitoyl-carnitine, acNP treatment does not significantly increase respiratory capacity (Fig. 7J-K). Thus, these data suggest that pyruvate oxidation and/or Complex I activity are specifically increased by acNP treatment in HFD mice. Hence, we analyzed Complex I-specific activity in frozen mice liver tissues. Interestingly, acNP treatment in LFD mice shows a trend to increase complex II activity, but not changes in Complex I activity (Supplementary Fig. 8G, H). However, the activity of Complex I increases in HFD mice treated with acNPs, recapitulating the results observed in fresh mice liver lysates (Fig. 7I). Overall, these data suggest that the functional quality of the mitochondria may be improved by acNPs, rather than total content. To assess the effects on liver inflammation induced by HFD and acNP treatment, we measured the expression of inflammatory markers, such as tumor-necrosis factor (TNF-α) and other cytokines and markers in total liver lysates. acNP treatment significantly decreases *TNF-α* levels as well as a trend to decrease other markers (Supplementary Fig. 8I).

Primary human hepatocytes are a favorable short-term human in vitro liver model because of their high functionality relative to the human organ in vivo[56]. Therefore, we tested whether acNPs protect from lipotoxicity in this model as well. As a model of lipotoxicity, we exposed primary human hepatocytes to 400 μM palmitate complexed to BSA at 4:1 ratio. Palmitate reduces maximal respiratory capacity without changes in mitochondrial mass, and acNP treatment reverses the decrease in respiratory capacity (Supplementary Fig. 8J, K). Specifically, the maximal respiratory rate (MRR) of mitochondria

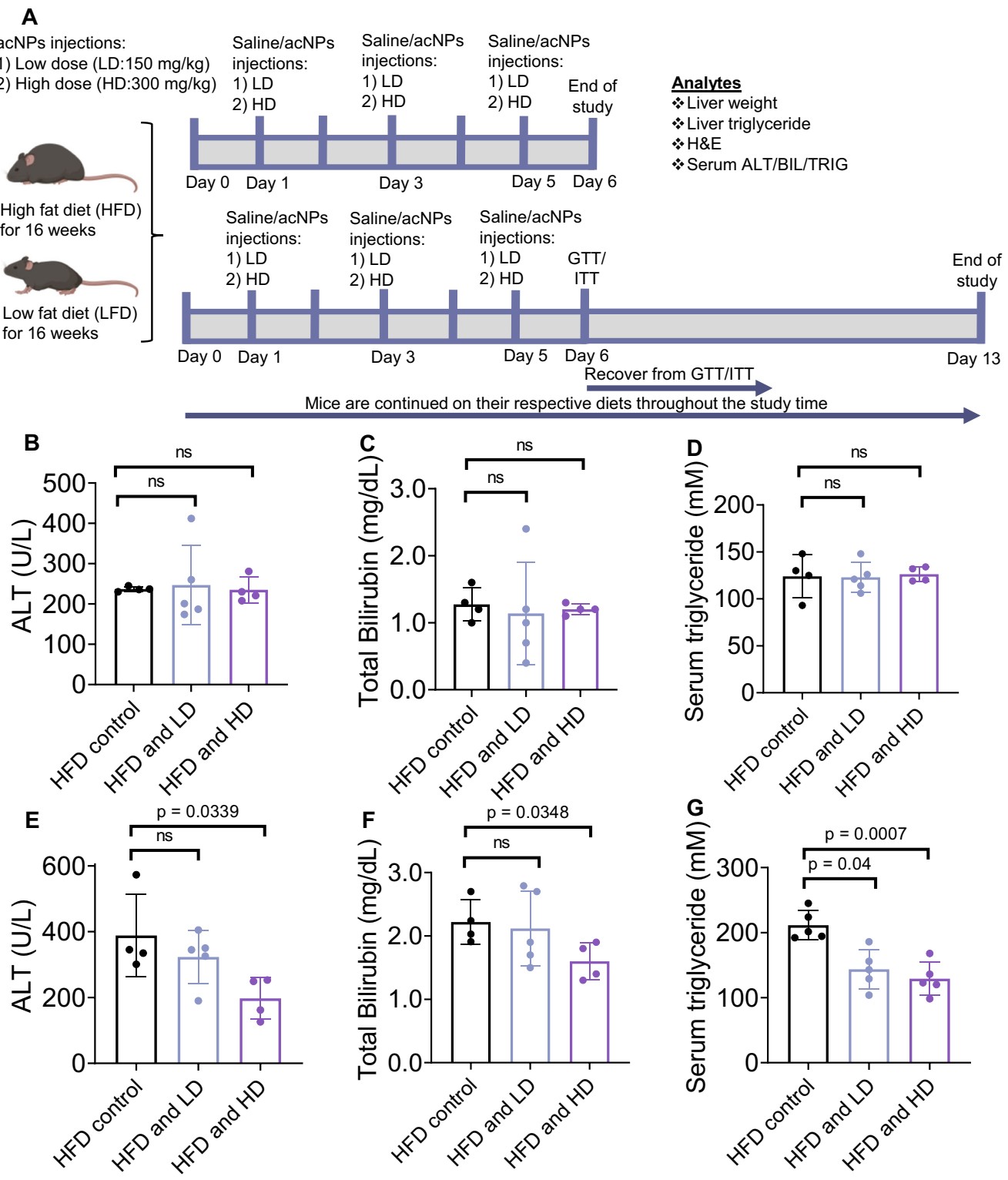

**Fig. 4 | AcNPs counteracts high-fat diet induced liver damage. A** Time chart of animal experiment with tail vein injection of acNPs. This schematic is created with BioRender.com<http://BioRender.com>. **B–D** Liver ALT, BIL levels and triglyceride levels in HFD-fed C57BL/6J mice serum after a single injection of LD or HD acNPs or (**E–G**) multiple injections of LD or HD acNPs for 6 days (*n* = 4 animals for HFD control, *n* = 5 animals for HFD and LD, *n* = 4 animals for HFD and HD). ALT and BIL levels do not change significantly as compared to control after single injection of either LD or HD acNPs, but decrease after multiple injections of HD acNPs, indicating no significant toxicity caused by acNPs. Serum triglyceride levels are decreased after multiple injections of LD or HD acNPs, indicating functional effect of acNPs in reducing serum triglyceride levels. Two-tailed unpaired *t*-test (**A–G**); data are expressed as means ± SD. n.s. not statistically significant. Source data are available as a Source Data file.

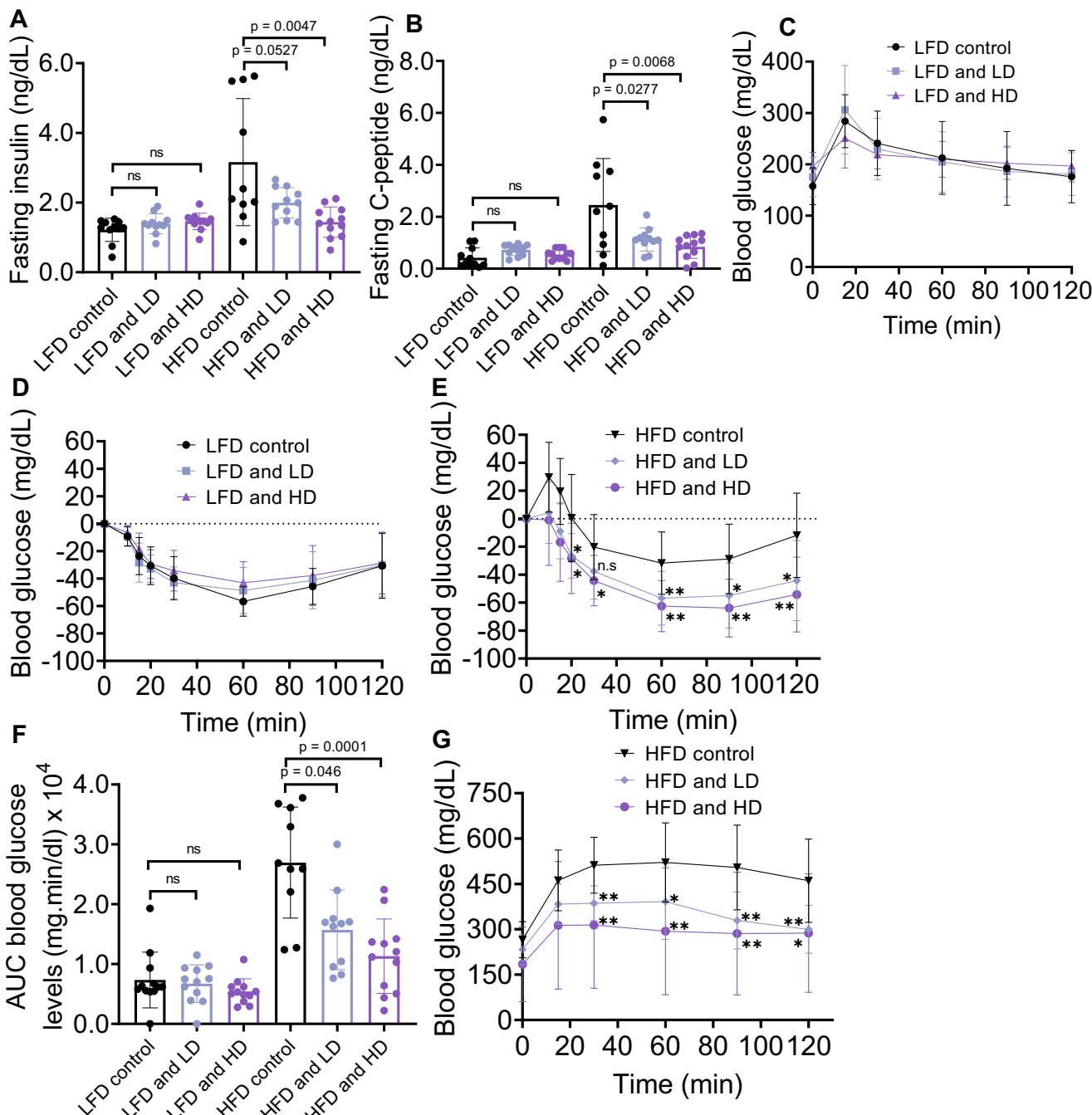

**Fig. 5 | Treatment with acNPs improves insulin sensitivity in high-fat diet-fed mice. A** Fasting insulin levels in mice serum. **B** Fasting c-peptide levels in mice serum. **C** Blood glucose response curves for low-fat diet (LFD) mice and **D** Insulin tolerance response curves for LFD mice. **E** Blood glucose response curves for high-fat diet (HFD) mice. Exact *p*-values between different treatment conditions are as follows: *p* = 0.0137 between HFD control and HFD and LD for 20 min time-point, *p* = 0.0204 between HFD control and HFD and HD for 20 min timepoint. *p* = 0.0643 between HFD control and HFD and LD for 30 min timepoint, *p* = 0.0101 between HFD control and HFD and HD for 30 min timepoint. *p* = 0.0074 between HFD control and HFD and LD for 60 min timepoint, *p* = 0.0013 between HFD control and HFD and HD for 60 min timepoint. *p* = 0.0143 between HFD control and HFD and LD for 90 min timepoint, *p* = 0.0011 between HFD control and HFD and HD for 90 min timepoint. *p* = 0.0131 between HFD control and HFD and LD for 120 min timepoint, *p* = 0.0015 between HFD control and HFD and HD for 120 min timepoint. **F** Area under curve (AUC) of ipGTT. High-dose acNPs (HD) show greater

improvement in blood glucose response compared to HFD control. **G** Insulin tolerance response curves for HFD mice. Exact *p*-values between different treatment conditions are as follows: *p* = 0.006 between HFD control and HFD and LD at 30 min timepoint, *p* = 0.0068 between HFD control and HFD and HD at 30 min timepoint. *p* = 0.0212 between HFD control and HFD and LD at 60 min timepoint, *p* = 0.0042 between HFD control and HFD and HD at 60 min timepoint. *p* = 0.0016 between HFD control and HFD and LD at 90 min timepoint, *p* = 0.0057 between HFD control and HFD and HD at 90 min timepoint. *p* = 0.0020 between HFD control and HFD and LD at 120 min timepoint, *p* = 0.0199 between HFD and HFD and HD at 120 min timepoint. For **A**–**G**, *n* = 11 animals for LFD control group, *n* = 11 animals for LFD and LD group, *n* = 12 animals for LFD and HD, *n* = 10 animals for HFD control group, *n* = 11 animals for HFD and LD group, *n* = 12 for HFD and HD group. Two-tailed unpaired *t*-test (**A**–**G**); data are expressed as means ± SD. *\*p* < 0.05, \*\**p* < 0.01, \*\*\**p* < 0.001, \*\*\*\**p* < 0.0001, n.s. not statistically significant. Source data are available as a Source Data file.

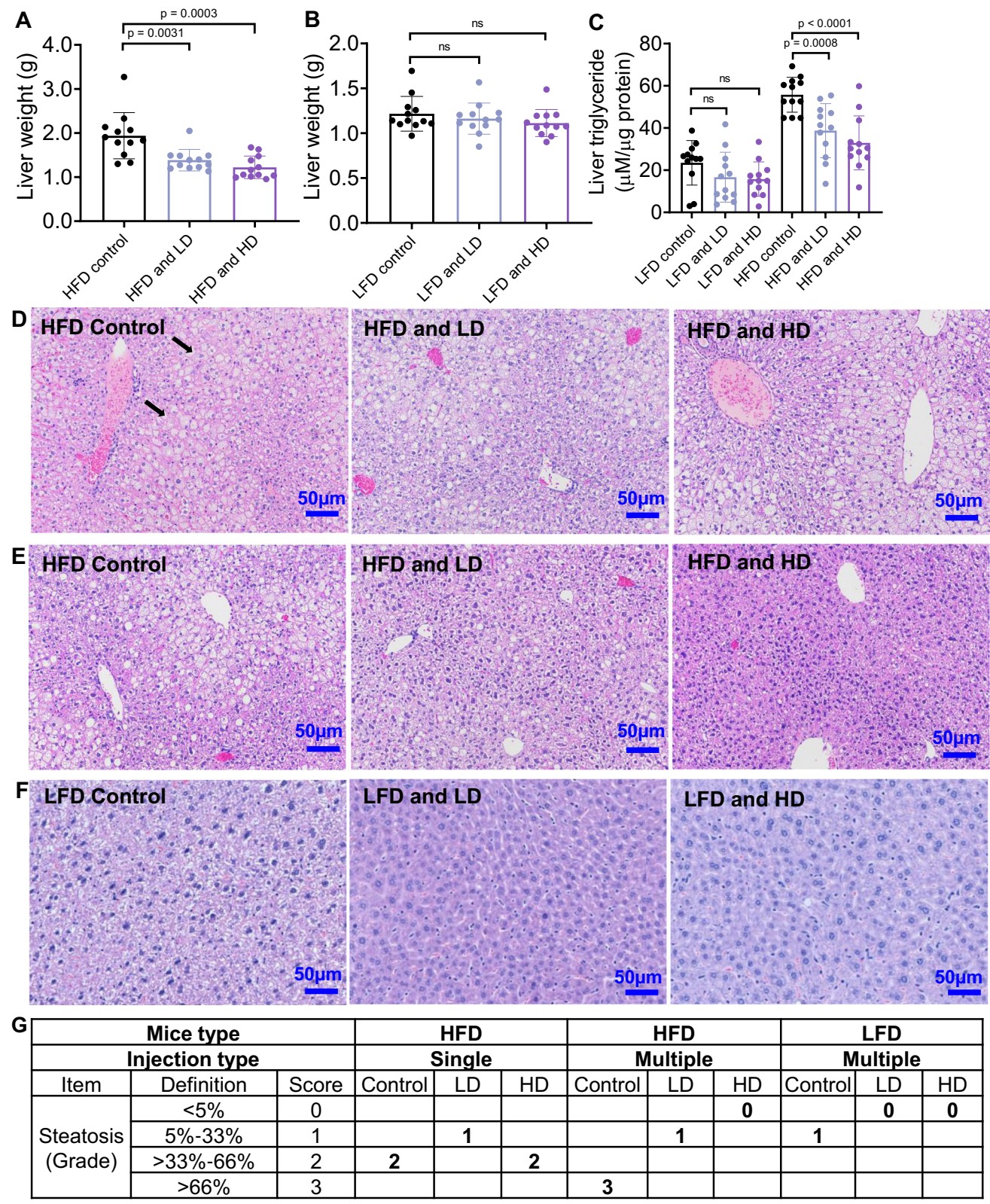

Fig. 6 | Treatment with acNPs reverses liver steatosis in high-fat diet-fed mice.
**A** Multiple injections of LD or HD acNPs to mice result in significant liver weight reduction, with more significant reductions with HD acNPs ($n = 12$ animals per group). **B** Multiple injections of LD or HD acNPs to LFD mice do not result in significant liver reduction ($n = 12$ animals per group). **C** Liver triglycerides level measured in mice across all treatment conditions ($n = 12$ animals per group). LD and HD acNPs result in liver triglyceride reduction, with HD acNPs having a greater response. Two-tailed unpaired $t$-test (**A**, **B**, **C**); data are expressed as means ± SD. n.s. not statistically significant. **D**–**F** Representative H & E stains of mice liver tissue slices from different treatments. Bar, 50 μm. Black arrows indicate lipid droplets or microvesicular steatosis ($n = 3$ animals were used for each treatment condition). **G** The degree of steatosis is determined with a histopathological grid (Supplementary Table. 3) and scored blindly by a pathologist. Source data are available as a Source Data file.

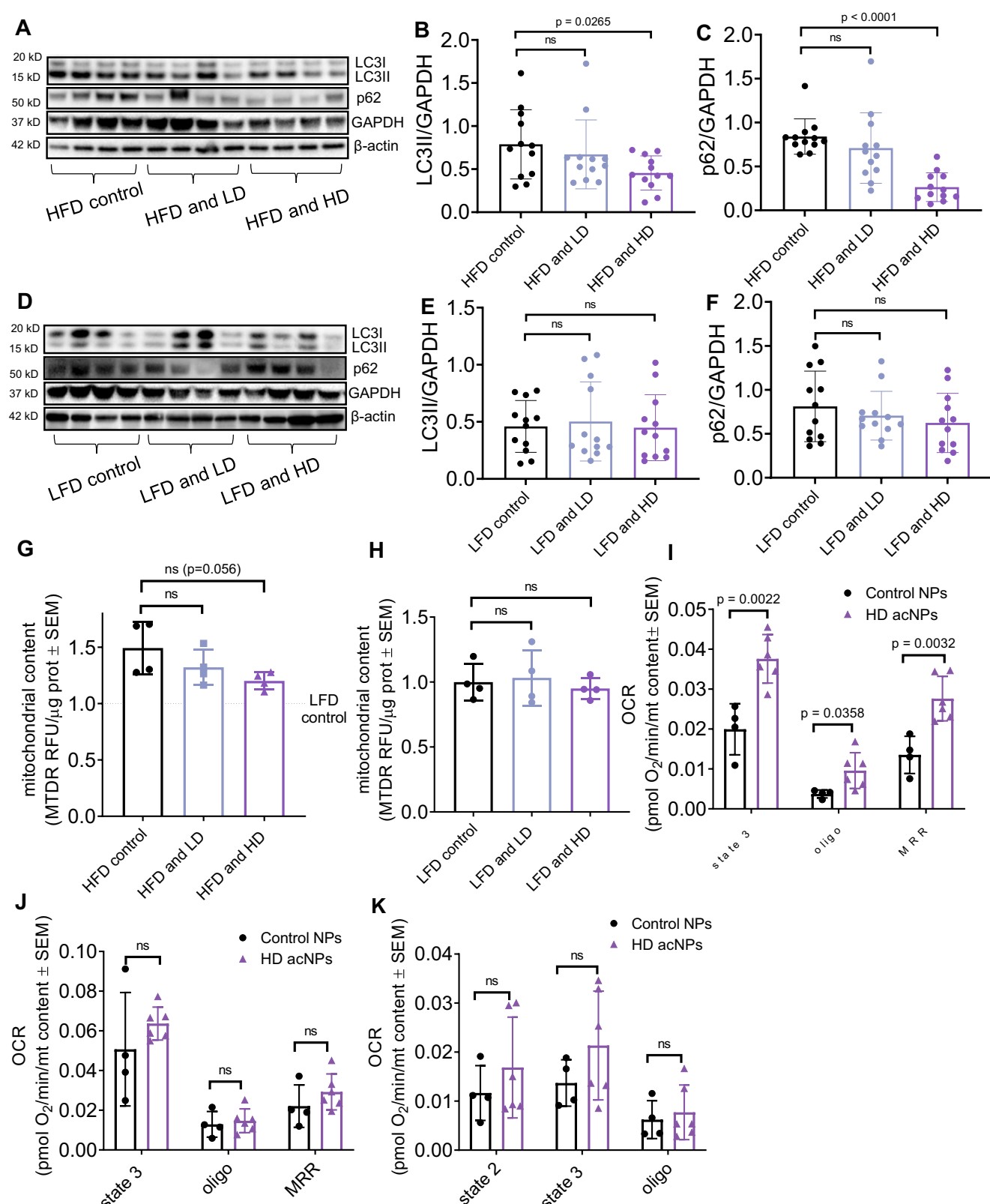

decreases in cells cultured in the presence of palmitate, while the addition of acNPs preserves MRR (Supplementary Fig. 8K). To further assess the possibility that lysosomal alkalization alone impairs mitochondrial maximal bioenergetics capacity, we induced lysosomal alkalization using bafilomycin A1 and measured OCR. Mitochondria respiratory capacity decreases upon bafilomycin A1 treatment, while acNP treatment restores OCR (Supplementary Fig. 8L). These data

indicate that the effect of acNPs to restore mitochondrial respiratory function is due to improvement of lysosomal acidification. Next, to determine the effect of lysosomal acidification on lysosomal homeostasis, we treated primary human hepatocytes with bafilomycin A1 alone or in combination with acNPs, and measured lysosomal enzyme activity using the Magic Red cathepsin L probe. Magic Red fluorescence intensity increases as the lysosomal cathepsin L activity

**Fig. 7 | AcNPs improve autophagic flux and mitochondrial oxidative capacity in high-fat diet-fed mice. A**–**C** Representative western blot exposure images showing protein expression of LC3II and p62 in HFD mice (*n* = 12 animals per treatment group). **D**–**F** Representative western blot exposure images showing protein expression of LC3II and p62 in LFD mice (*n* = 12 animals per treatment group). **G** Mitochondria content of mice livers under HFD, HFD and LD acNP, and HFD and HD acNP treatments (*n* = 4 animals per treatment group). **H** Mitochondria content of mice livers under LFD, LFD and LD acNP, and LFD and HD acNP treatments (*n* = 4

animals per treatment group). **I** Oxygen consumption rate of mice liver lysates mitochondria in pyruvate-malate substrate (*n* = 4 animals in control NPs group, *n* = 6 animals in HD acNPs group). **J** Succinate-rotenone substrate (*n* = 4 animals in control NPs group, *n* = 6 animals in HD acNPs group), and **K** Palmitoyl-L-carnitine substrate (*n* = 4 animals in control NPs group, *n* = 6 animals in HD acNPs group). Oligo: Oligomycin, MRR: Maximal respiratory rate. Two-tailed unpaired *t*-test (**B**, **C**, **E**, **F**, **G**, **H**, **I**, **J**, **K**); data are expressed as means ± SD. n.s. not statistically significant. Source data are available as a Source Data file.

increases (Supplementary Fig. 8M, N). Treatment with either bafilomycin A1 or palmitate decreases Magic Red intensity, indicating decreased lysosomal enzyme activity. Addition of HD acNPs preserves Magic Red intensity, but not with LD acNPs, indicating that the increase in lysosomal activity is dose dependent due to changes in lysosomal acidification.

## Discussion

Previous strategies to increase autophagic flux to counteract NAFLD used mTOR inhibitors like rapamycin to increase autophagic turnover, which in turn increased V-ATPase expression and lysosomal acidification. However, there are currently no specific strategies to directly target and increase lysosomal acidification (reduce pH) and reverse hepatic steatosis. To induce a direct re-acidification of lysosomes, we synthesized acid-activated nanoparticles (acNPs) of around 100 nm in diameter for rapid endocytic uptake and localization to lysosomes[57]. We designed the acNPs to contain ester linkages that degrade around pH 6, the pH of a dysfunctional lysosome, and release tetrafluorosuccinic acid. This acid readily deprotonates in the lysosome to increase the overall proton concentration and hence lower lysosomal pH. The re-acidification of the lysosomes in HepG2 cells under lipotoxicity rescues cathepsin L lysosomal enzyme activity and autophagic flux, which lead to a reduction in lipid accumulation. In high-fat diet-fed mice, the treatment with acNPs restores autophagic flux and mitochondria function, reduces lipid droplet accumulation, and promotes insulin sensitivity. Mechanistically, we propose that acNPs reverse NAFLD by promoting the elimination of free fatty acids, by restoring lipophagy and mitochondrial energy expenditure due to the rescue of lysosomal function and autophagic flux.

A few research groups, including ours, have synthesized other types of acidifying nanoparticles, such as the photo-activated nanoparticles (paNPs)[39] and poly (lactic-co-glycolic) acid nanoparticles (PLGA NPs)[34,58], which upon treatment in vitro reduce lysosomal pH in a type II diabetes (lipotoxicity induced) and a neurotoxin induced Parkinson's disease model. The paNPs require application of an external UV-light trigger, rendering them unsuitable for many in vivo models. The PLGA NPs, on the other hand, only slightly lower the lysosomal pH. Upon comparison of the two types of NPs, acNPs afford a 4 times more significant lysosomal pH reduction than PLGA NPs in HepG2 cells under palmitate treatment (Supplementary Fig. 9A). Furthermore, acNPs significantly reduce the lipid droplets number in HepG2 cells more than PLGA NPs, due to the higher acidification capability of acNPs, providing a greater functional rescue (Supplementary Fig. 9B).

In sum, impaired lysosomal acidification results in decreased autophagic flux and cellular function, and is implicated in several pathologies including NAFLD, type 2 diabetes[39,54,59] and neurodegenerative diseases[58,60]. Consequently, new biological tools to control lysosomal pH and treatments to restore normal autophagic processes and cell function are of keen interest. The key to the activity of the acNPs lies in its size and negative zeta potential, which ensures rapid hepatocyte uptake and localization in the lysosome, together with its polyester composition, which liberates acid to lower lysosomal pH upon lysosome entry. Hence, the acNPs are a

responsive pharmacological therapy, which rely on a biological cue for activation. Unlike the majority of NP-based drug delivery systems where the NP is delivering an active agent, the acNP itself is the active agent or biomaterial. Accordingly, targeting lysosomal acidity with acNPs represents a novel strategy to improve lysosome function and autophagic flux in hepatocytes and holds potential for treating lysosome dysfunction where impaired autophagic flux contributes to the pathology.

## Methods

### Ethics statement

C57BL/6J DIO control (male, 16 weeks) and C57BL/6J DIO control (male, 16 weeks), FVB/NJ and C57BL/6N (male, 17 weeks) were used in this study. The mice were housed in a temperature-controlled room (25 °C) in virus-free facilities on a 12-h light/dark cycle (7:30 a.m. on/ 7:30 p.m. off) and water *ad libitum*. Experimental procedures conducted on mice were performed in accordance with animal welfare and in compliance with other related ethical regulations. The mice studies were conducted under an approved Institutional Animal Care and Use Committee (IACUC) protocol at Boston University, under animal protocol number 17-014, and under the IACUC protocol at University of California, Los Angeles, under animal protocol number ARC-2017-019.

### General methods, materials, and instrumentation

Materials and chemicals were purchased from Sigma Aldrich, Thermofisher Scientific and Biosynth, and were used as received unless otherwise noted. Solvents used for the polymerization reactions were dried prior to use. All reaction flasks were oven dried overnight before use. $^1$H-NMR, $^{13}$C-NMR and $^{19}$F-NMR spectra were recorded on a Varian INOVA 500 MHz spectrometer and CDCl$_3$ was used as solvent. Polymer molecular weights were determined by gel permeation chromatography (GPC) versus monodisperse polystyrene standards using THF as the eluent at a flow rate of 1.0 mL/min through two Jordi columns (Jordi Gel DVB 10$^5$ Å and Jordi Gel DVB 10$^4$ Å, 7.8 × 300 mm) at 25 °C in series with a refractive index detector.

### Polymers synthesis

25% poly(ethylene tetrafluorosuccinate-co-succinate) (PEFSU) polymer synthesis: Di-acid monomers tetrafluorosuccinic acid (Thermofisher Scientific, Cat # T16215G), succinic acid (Sigma Aldrich, Cat # 398055) were added at a 1:3 molar ratio in a round bottom flask. Ethylene glycol (Sigma Aldrich, Cat # 324558) were added at 10 mol% excess, together with metal catalyst titanium isoproxide (TIPT) (Sigma Aldrich, Cat # 205273), and distilled azetropically at 120 °C for 16 h. Subsequently, a vacuum of 100 mtorr was slowly applied to prevent excessive foaming and to minimize oligomer sublimation in the reaction vessel and undergo further condensation to form higher molecular weight polymer chains. The temperature was also increased to 130–140 °C for at least 12 h. Finally, the crude product is precipitated in cold diethyl ether (Thermofisher Scientific, Cat # AC615085000) and dried under high vacuum for storage and further use. Poly(ethylene succinate) (PESU) polymer is synthesized in a similar distillation method as 25% PEFSU, except that only succinic acid is added to the round bottom flask in the first step.

## Nanoparticles synthesis and characterization

The control and acNPs are formed from PESU and 25% PEFSU polymer, respectively, using nanoprecipitation. For synthesis of control NPs, 50 mg PESU is dissolved in 500 μL dichloromethane (DCM) (Sigma Aldrich, Cat #270997) and added to a solution of 10% wt/vol sodium dodecyl sulfate (SDS) (Sigma Aldrich, Cat #436143) dissolved in 2.0 mL of phosphate buffer saline or medical grade saline (Fisher Scientific, Cat # Z1376). The organic and aqueous phase were then combined and sonicated for 10 min at 80 W with a 1 s pulse, 2 s delay under argon atmosphere to create an emulsion. After sonication, the emulsion is stirred to evaporate off excess DCM and dialyzed against Milli-Q water for 24 h using a SnakeSkin dialysis tubing (MWCO 10KDa) (Thermofisher Scientific, Cat # 68100). For synthesis of acNPs, 50 mg of 25% PEFSU polymer is dissolved in 0.5 mL of acetonitrile (ACN) (Sigma Aldrich, Cat #271004) and filtered through a 0.2 μm syringe filter (Millipore, Cat # Z741696) to remove large particulates. SDS is dissolved in phosphate buffer saline or medical grade saline (Fisher Scientific, Cat # Z1376) and filtered through a 0.2 μm syringe filter (Millipore, Cat # SLFG033). The solution of polymer in ACN is then added drop wise into the fast stirring SDS aqueous solution. Immediately after, the emulsion is placed into a SnakeSkin dialysis tubing (MWCO 10KDa) (Thermofisher Scientific, Cat # 68100) and dialyzed against Milli-Q water for 24 h. For Dynamic Light Scattering (DLS) measurements, 200 μL of the solution is diluted in 2.8 mL of DI water, and the size and zeta potential are obtained from the Brookehaven dynamic light scattering instrument (Nova Instruments Rehovot Israel). All measurements were performed in triplicates.

## Nanoparticles pH titration

Control NPs and acNPs were diluted in 20 mM pH 7.4 or pH 6.0 phosphate saline buffer (10 mg/mL). Nanoparticles were stirred and the pH measured at intervals using a pH meter. All measurements were performed in triplicates.

## Gel permeation chromatography (GPC) degradation assay

Control NPs and acNPs were added to a pH 6.0, 20 mM PBS buffer at a final concentration of 10 mg/mL. At each timepoint, an aliquot of NP stock was flash frozen in liquid nitrogen, and lyophilized. For processing on GPC, the samples were dissolved in 0.7 mL tetrahydrofuran (THF), filtered with a 0.22 μm syringe filter, and run against a narrow polystyrene standard curve (Aligent, PS-1, Standard B) at a flow rate of 1 mL/min in THF.

## Rhodamine-labeled-acNPs (Rho-acNPs) synthesis and characterization

Rho-acNPs polymer synthesis was carried out following a procedure as described[61]. 25% PEFSU polymer was covalently coupled with a fluorescent dye, rhodamine B. Briefly, rhodamine B (Sigma Aldrich, Cat #83689) and 4-(Dimethylamino)pyridine (DMAP) (Sigma Aldrich, Cat # 522805) were dissolved in dry dichloromethane at room temperature under argon atmosphere. After 40 min of stirring, 1-(3-Dimethylaminopropyl)−3-ethylcarbodiimide hydrochloride (EDCI. HCl) (Biosynth, Cat # FD05800) dissolved in dry dichloromethane was added to the reaction medium cooled in an ice bath. After another 40 min under stirring, 25% PEFSU polymer dissolved in dry dichloromethane was added. The reaction medium was kept under stirring for 2 days in an argon atmosphere at room temperature. The extraction was carried out with aqueous solutions of saturated sodium bicarbonate and excess water removed with brine. The fluorescent product was then purified through silica gel column chromatography using dichloromethane/methanol as the elution solvent.

## Scanning electron microscopy

NPs were diluted 1: 200 or 1:500 times in Milli-Q water. Aliquots were plated on silicon wafers and allowed to air dry overnight. The wafers were then affixed to aluminum stubs with copper tape and sputter coated with 5 nm Au/Pd. These samples were then imaged using a Supra 55VP field emission scanning electron microscope (ZEISS) with an accelerating voltage of 2 kV and working distance of 5.5 cm. Image J (Fiji) was used to quantify nanoparticle sizes and the results are analyzed using GraphPad Prism v9.

## Palmitate: BSA preparation

Palmitate was first dissolved in DMSO (Millipore, Cat # 41639), and subsequently this solution was dissolved at 45 °C in DMEM media (no glucose) containing 6.7% fatty acid-free BSA (EMD Millipore, Cat # 126609) to make a 4 mM (10×) stock. For control BSA conditions, a 10× stock of DMEM media containing 5% BSA and 1% DMSO was used. For the treatment conditions, the 10× stocks were added to DMEM media containing 5% FBS, 50 U/mL penicillin, and 50 g/mL streptomycin. The pH of the treatment media was then adjusted to 7.4, and the adjusted media was sterile filtered using a 0.45 μm syringe filter (Millipore, Cat # SLHV004SL) before treating the HepG2 cells for 16 h.

## Cell culture and cytotoxicity

HepG2 cells (ATCC, Cat # HB-8065) were cultured in DMEM media supplemented with 10% FBS, 1 mM glutamine, 50 units/mL penicillin, and 50 g/mL streptomycin. The cytotoxicity of acNPs with and without palmitate was evaluated using an MTS cell proliferation assay (Abcam, Cat # ab197010). HepG2 cells were cultured in a 96-well plate at 15,000 cells/well for 1 day, after which the media was exchanged for media containing no treatment or 400 μM palmitate or 0, 50, 250, 500 or 1000 μg/mL of acNPs. The cells were then incubated with treatment for 24 h, after which cell viability was quantified with a Biotek Synergy HT plate-reader relative to the no treatment control, after correcting for background absorbance.

## Flow cytometry

FACS analyses of rhodamine-labeled acNP-treated cells was done with 620 FACScan. FACS data analysis was performed using FACScalibur (Beckman Coulter). $5 \times 10^5$ cells per timepoint were trypsinized, washed twice with PBS by centrifugation, and then subjected to flow cytometry. Cell debris was excluded by gating on the forward and side scatter plot. Images of gating strategy can be found in Supplementary Fig. 2H.

## Lysosensor staining and image analysis

For co-localization imaging, cells were first incubated with Rho-acNPs for 24 h and LysoTracker blue dye (Thermofisher Scientific, Cat # L7525) added according to manufacturer's protocol for 2 h. The cells are incubated with fresh media and imaged using confocal microscopy. Cells were stained with 1 μM LysoSensor yellow/blue (Thermofisher Scientific, Cat # L7545) for 5 min followed by confocal imaging using a 360 nm excitation and collecting images at the yellow wavelength range (510–641 nm) and at the range of blue wavelength (404–456 nm) using the Zeiss LSM 880 Confocal Microscope. The ratio between yellow and blue was calculated using Metamorph software. In brief, background noise was removed by a median filter, followed by thresholding to identify individual lysosomes. Mean yellow and blue fluorescence intensities were obtained for the identified lysosomes, and yellow/blue ratio values were calculated. Quantification of pH changes was achieved by imaging Lysosensor fluorescence in 2-(N-morpholino) ethanesulfonic acid buffer of varying pH and establishing a standard curve of Lysosensor fluorescence ratio to pH. For representative images shown, ratio images were generated by dividing yellow and blue Lysosensor images.

## Magic red cathepsin L activity assay

HepG2 cells were stained with 10 μg/mL Magic red cathepsin L (MR-cathepsin L; Immunochemistry Technologies, Cat # 941) for 1 h. The

cells were then washed three times with PBS and imaged using either confocal microscope or fluorescence plate-reader at 531/40 excitation; 629/53 emission.

## HepG2 and primary human hepatocytes bioenergetics

HepG2 and primary human hepatocytes were plated in a seahorse plate at $6 \times 10^3$ cells per well 24 h before treatments were started. Cell were treated with Bafilomycin A1 (200 nM) (Thermofisher Scientific, Cat # SML1661) or Bafilomycin A1 plus acNPs (at concentrations from 30 µg/mL to 200 µg/mL) for 5 h. Cells were incubated under glucolipotoxicity (GLT, 150 µM BSA-palmitate plus 25 mM glucose) or GLT plus acNPs (30 µg/mL to 200 µg/mL) for 18 h. Respiration was assessed using a Seahorse bioanalyser in an assay media containing 10 mM glucose (Sigma Aldrich, Cat # G7021), 2 mM pyruvate (Sigma Aldrich, Cat # P76225) and 4 mM glutamine (Sigma Aldrich, Cat # G3126). Injections were as follows: oligomycin in port A (2 µM final concentration) (EMD Millipore, Cat # 495455), FCCP (in bafilomycin treated cells) (Enzo Life Sciences, Cat # BML-CM120-0010) or BAM 15 (in GLT treated cells) (Cayman Chemical Company, Cat # 17811) in port B (1.5 µM or 10 µM) and antimycin A in port C (2 µM) (Enzo Life Sciences, Cat # ALX-380-075-M010). Port injection mix were prepared in assay media but for BAM 15 that was prepared in 600 µM oleate, 100 µM glutamine and 1 mM pyruvate. Cells were fixed after the assay and stained with DAPI. Cell counts were assessed using the Operetta system.

## Insulin sensitivity assay

HepG2 cells cultured in DMEM (5 mM glucose) supplemented with 10% FBS, and 1 mM glutamine were plated into 6 well plates at 300,000 cells per well 24 h before the treatments were started. Cells were incubated with BSA, BSA + acNPs, Palmitate, or Palmitate + acNPs for 18 h. Palmitate complexed to BSA at 4:1 ratio was used at 150 µM and acNPs at 100 µg/mL. Then, cells were deprived of FBS for 6 h, treated with 10 nM insulin (Humulin, Humulin R U-100) for 10 min, washed twice with PBS, and frozen till the extraction with RIPA lysis buffer (Thermofisher Scientific, Cat # 89900) for immunoblotting.

## Western blot

Cell samples were washed with PBS twice on ice, followed by lysing with RIPA buffer (Thermofisher Scientific, Cat # 89900) containing an additional 2% Triton-X-100, protease and phosphatase inhibitors (Thermofisher Scientific, Cat # 78440). The collected lysates were kept on ice for 15–30 min followed by centrifugation for 10 min at 13,500 × g at 4 °C. The supernatants are subjected to Pierce™ BCA Protein Assay Kit (Thermo Fisher Scientific, Cat # 23225) to determine total protein concentrations.

For mice tissue samples, the mice were euthanized at the end of experiment, and tissues were snap-frozen in liquid nitrogen and minced via mortar and pestle, before storing at −80 °C. For western blot protein samples preparation, frozen tissue powders were homogenized in RIPA lysis buffer (Thermofisher Scientific, Cat # 89900) containing 1% SDS, 2% triton X-100 (Sigma Aldrich, Cat # X-100), and protease and phosphatase inhibitors (Thermofisher Scientific, Cat # 78440). The homogenates are then centrifuged, and the supernatants were collected. Samples were loaded in 4–12% polyacrylamide gel (Invitrogen, Cat # NP0323BOX) and transferred onto a polyvinylidene difluoride membrane (Invitrogen, Cat # LC2002) using a wet (tank) transfer machine. LC3A/B (Cell Signaling, Cat # 12741; RRID: AB_2617131, 1:1000), GAPDH (Cell Signaling, Cat # 2118; RRID: AB_561053, 1:1000), β-actin (Cell Signaling, Cat # 4967; RRID: AB_330288, 1:1000), SQSTM1/p62 (Cell Signaling, Cat # 5114; RRID: AB_10624872, 1:1000), Akt (Cell Signaling, Cat # 4685; RRID: AB_2225340, 1:1000), p-Akt (Ser 473) (Cell Signaling, Cat # 4060; RRID: AB_2315049, 1:1000), Insulin Receptor β (IR) (Cell Signaling, Cat # 3025; RRID: AB_2280448, 1:1000), p-IR Tyr1162/1163 (Invitrogen, Cat # 44–804; RRID: AB_2533762, 1:1000), GSK-3β (Cell Signaling, Cat # 12456; RRID: AB_2636978, 1:1000), p-GSK3β (Ser 9) (Cell Signaling, Cat # 9336; RRID: AB_331405, 1:1000), Total OXPHOS Rodent WB Antibody Cocktail (Abcam, Cat # ab110413; RRID: AB_2629281, 1:1000), or Vinculin (Sigma Aldrich, Cat # V9131; RRID: AB_477629, 1:1000) antibodies were incubated overnight, and Anti-rabbit IgG, HRP-linked Antibody (Cell Signaling, Cat # 7074; RRID: AB_2099233, 1:3000), or Anti-mouse IgG, HRP-linked Antibody (Cell Signaling, Cat # 7076; RRID: AB_330924; 1:3000) was used as secondary antibodies for detection of bands. Densitometry was performed with Fiji Image J v.1.5.3 s and protein expression levels were normalized to GAPDH, β-actin or Vinculin.

## Animal handling

All procedures were conducted with the approval of the Institutional Animal Care and Use Committee of the Boston University and in accordance with National Institutes of Health Guidelines for the Care and Use of Laboratory Animals. We used an established in vivo model of NAFLD mice[62–64] where C57BL/6J DIO male mice (Jackson Laboratory, Strain # 380050) fed with high-fat diet (HFD) (D12492: 60% kcal energy as fat, Research Diets, Inc) for 16 weeks was chosen as the in vivo model of NAFLD, and C57BL/6 J DIO control male mice (Jackson Laboratory, Strain # 380056) on low-fat diet (LFD) (D12450J: 10% kcal energy as fat, matching sucrose for D12492) for 16 weeks were used as the healthy mice control. We only chose male mice because female mice have been reported to be protected against high-fat diet-induced metabolic syndrome[65]. The mice were housed in a temperature-controlled room (25 °C) in virus-free facilities on a 12-h light/dark cycle (7:30 a.m. on/7:30 p.m. off) and water *ad libitum*. In each animal experiment, mice were randomly allocated to receive tail vein injections of either saline injection or with low dose or high-dose acNPs. During the injection time period, we monitored the mice body weights and food intake for initial signs of toxicity. Food intake were measured every 3 days, and body weights were measured weekly. All mice were maintained under specific pathogen-free conditions and treated with humane care under approval from the IACUC protocol 17-014 from Boston University, as well as the IACUC protocol ARC-2017-019 from University of California, Los Angeles.

## Biodistribution study

All procedures were performed using pre-chilled equipment and solutions. Biodistribution studies were done in FVB/NJ (Jackson Laboratory, Strain # 001800) and C57BL/6N mice (Jackson Laboratory, Strain # 005304). In FVB/NJ mice, saline (vehicle) or Rho-acNPs were injected three times (one injection every other day) into the mice. The organs (e.g., liver, kidney, brain, white adipose tissue, heart, lungs) were harvested following 48 h of the last injection and rinsed in PBS. The tissues were weighed, minced, and suspended in 6 mL of PBS. The preparation was then mechanically homogenized with 9 strokes in glass-Teflon dounce homogenizer. The total protein content from each tissue was measured, and 2 µg, 10 µg, 20 µg, and 100 µg of the protein from different tissues was used to assess for linear range of rhodamine fluorescence using Tecan 2000 Plate-reader at (Ex/Em: 553/627 nm). For the comparison between different organ tissues, 10 µg of protein per tissue were used, and fluorescence signal is expressed as fold change to saline control.

## Immunofluorescence

C57BL/6J or C57BL/6N mice tissue sections were stored in optimal cutting temperature (OCT) medium. Frozen samples were sectioned using a cryostat and allowed to air dry on the slide 10–15 min prior to fixation. Tissue sections were covered with ice-cold 100% methanol and incubated in methanol for 10 min at −20 °C, rinsed in PBS for 5 min. Tissue sections were incubated in permeabilization buffer (0.3% Triton X-100 and 0.5 mg/mL sodium deoxycholate in PBS, pH

7.4) for 15 min at room temperature. Subsequently, tissue sections were blocked with 0.3% Triton X-100 with 5% normal goat serum in PBS for 1 h at room temperature and then incubated with primary antibody of LAMP-1 (BioLegend, Cat # 121602; RRID: AB_572021, 1:200), or anti-CLEC4F (BioLegend, Cat # 156803; RRID: AB_2814081, 1:200) at 4 °C overnight. The next day, the cells were washed in PBS and incubated with Anti-Mouse Alexa Fluor 488 (Thermofisher Scientific, Cat # A11001; RRID: AB_2534069, 1:3000) for 1 h at room temperature. DAPI staining was added 15 min prior super-resolution microscopy imaging.

For Rho-acNPs and Kupffer cells co-localization analysis, liver tissue sections were analyzed with QuPath software v.0.2.3. to determine percent and area of CLEC4F and Rhodamine-positive cells. An average of 60000 cells were counted per slide.

### Blood and serum analysis

A blood sample from each mouse was collected via cardiac puncture into a SST vacutainer blood collection tube (Thermofisher Scientific, Cat # 15-350-30). Serum was separated by centrifugation at $10,000 \times g$ at room temperature for 15 min. Blood hematological parameters, and serum alanine aminotransferase (ALT), bilirubin (BIL), and triglycerides levels were measured using the Hemavet and IDEXX chemistry analyzer, respectively, at the Boston University Medical Center (BUMC) Analytical Core.

The serum analysis was replicated in University of California, Los Angeles (UCLA), using LFD-fed mice, HFD-fed C57BL/6N[49] with 17 weeks of HFD. Serum alkaline phosphatase (ALP), alanine aminotransferase (ALT), aspartate aminotransferase (AST), creatine kinase (CK), total bilirubin (BIL), albumin levels were measured at the Pathology & Laboratory Medicine Services at UCLA.

### Glucose tolerance and insulin tolerance tests

Glucose and insulin tolerance tests were performed on mice fasted for 6 h in the morning. Glucose (2 g/kg body weight) or insulin (0.75 U/kg body weight) was administered via intraperitoneal injection. Blood glucose concentrations were measured using blood glucose strips with a blood glucometer (CVS Advanced Health). Glucose measurements were performed at baseline and 15, 30, 60, 90, and 120 min after glucose injection. Blood was collected from the tail in Microvette® CB 300 K2E capillary tubes (Sarstedt Inc, Cat # NC9141704) at the first 0 and 15 min timepoint for the glucose and insulin tolerance tests and used for analysis of serum insulin and c-peptide levels. The area under the glucose tolerance curves were evaluated using the trapezoidal method with ORIGIN PRO 8 software.

To assess insulin action at the tissue level, mice fasted for 6 h were injected with insulin 0.75 U/kg for HFD mice and 0.50 U/kg for LFD mice intraperitoneally and blood glucose measurements were performed at baseline, and 10, 15, 20, 30, 60, 90, and 120 min later. The mice are sacrificed 1 week after glucose tolerance tests or insulin tolerance tests, and the liver weights measured. Ultra-Sensitive Mouse Insulin ELISA kit (Crystal Chem, Cat # 50-194-7920) and Mouse C-peptide ELISA kit (Crystal Chem, Cat # 80954) was used to quantify insulin and c-peptide in the plasma, respectively, according to manufacturer's instructions.

### Liver histochemistry

Liver tissues were fixed in 10% formaldehyde neutral buffer solution for 24–48 h, embedded in paraffin, and cut into thin slices. The liver sections were obtained from two mice per treatment group, and liver sections were stained with trichrome stain by the Collaborative Research Laboratory (CoRe) at Boston University Chobanian & Avedisian School of Medicine. The extent of steatosis and inflammation was reviewed by a board-certified veterinary pathologist in a blinded fashion and scored using the provided pathological scoring grid criteria[66].

### Liver lysates and mitochondrial isolation

The liver was removed, washed with PBS, minced with scissors. Samples were placed in a Potter-Elvehjem (Teflon-glass) homogenizer (6–9 strokes) with 10 mL of ice-cold isolation buffer (250 mM mannitol (Sigma Aldrich, Cat # M9546), 75 mM sucrose (Thermofisher Scientific, Cat # L-12686), 100 µM K-EDTA (Sigma Aldrich, Cat # 03660), 10 mM KHEPES (Corning, Cat # 25060CI, pH 7.4) supplemented with 500 µM K-EGTA (Sigma Aldrich, Cat # E4378, pH 7.4). Homogenates were centrifuged at $1000 \times g$ for 10 min at 4 °C, then the supernatant was removed and re-centrifuged at $1000 \times g$ for 10 min at 4 °C. We collected 1 mL of the resulting supernatant as the homogenate (lysate) sample and the rest was centrifuged at $10,000 \times g$ for 10 min at 4 °C. The mitochondrial pellets were washed twice in wash buffer (isolation buffer supplemented with 0.5% fatty acid-free BSA). The final mitochondrial pellet was re-suspended in ice-cold isolation buffer with no BSA. BSA was omitted from the final isolation buffer to prevent interference with the protein assay kit.

### Bioenergetics on lysates and isolated mitochondria

Mitochondria and lysates were loaded into Seahorse XF96 microplate in 20 µL of MAS containing substrates. The loaded plate was centrifuged at $2000 \times g$ for 5 min at 4 °C (no brake) and an additional 130 µL of MAS + substrate was added to each well. To avoid disrupting mitochondrial adherence to the bottom of the plate, add MAS using multichannel pipette pointed at a 45° angle to the top of well chamber, as instructed by the manufacturer. Substrate concentrations were as follow: (i) 5 mM pyruvate (Sigma Aldrich, Cat # P76225) + 5 mM malate (Sigma Aldrich, Cat # M7397) + 4 mM ADP (Sigma Aldrich, Cat # A5285) (PMA), (ii) 5 mM succinate (Sigma Aldrich, Cat # S9512) + 2 µM rotenone (Sigma Aldrich, Cat # R8875) + 4 mM ADP (SRA) or (iii) 2 mM malate + 4 mM ADP (MA). For PMA and SRA, injections were as follows: oligomycin at port A (3.5 µM), FCCP at port B (4 µM), N, N, N′, N′-Tetramethyl-p-phenylenediamine (TMPD) (Sigma Aldrich, Cat # 87890) + ascorbic acid (Thermofisher Scientific, Cat # A61-100) (0.5 mM + 1 mM) at port C and azide (50 mM) at port D. For MA injections were as follows: Palmitoyl-carnitine (Sigma Aldrich, Cat # P1645) + malate+ ADP (0.4 mM + 5 mM + 4 mM) at port A, oligomycin at port B (3.5 µM), N, N, N′, N′-Tetramethyl-p-phenylenediamine (TMPD) + ascorbic acid (0.5 mM + 1 mM) at port C and azide (50 mM) at port D (4 µM). Mix and measure times were 0.5 min and 4 min, respectively. A 2 min wait time was included for oligomycin-resistant respiration measurements.

For liver mitochondria, 6 µg for PMA and MA and 3 µg for SR-dependent respiration was used. For liver lysate, we loaded 15 µg for PMA and MA and 8 µg for SR-dependent respiration was used.

### Mitochondrial content quantification

For mitochondrial content quantification 15 µg of liver lysate was seeded per well on a clear-bottom black 96-well plate in 100 µL of MAS containing MitoTracker Deep Red FM (MTDR) (Thermofisher Scientific, Cat # M22426) (1 µM, final) and incubated at 37 °C for 10 min. Plates were centrifuged at $2000 \times g$ for 5 min at 4 °C (no brake) and supernatant was carefully removed. Finally, 100 µL of MAS was added per well and MTDR fluorescence measured as indicated before. Mitochondrial content was calculated at MTDR signal (minus blank) per milligram of protein.

### Seahorse data analysis

Wave software (Agilent) was used to export OCR rates normalized by protein (isolated mitochondria and lysates) or mitochondrial content (lysates) to GraphPad Prism v9.

### RNA isolation and gene expression

Total RNA was extracted from approximately 10 mg of flash frozen livers using Trizol® reagent (Ambion) to lyse the cells and

homogenized with the Powergen 125 polytron homogenizer. RNA is then collected via aqueous/organic phase separation and put through the Qiagen RNeasy Mini Kit RNA-binding column to purify the RNA according to manufacturer's instructions. RNA concentration was determined by Nanodrop (Thermofisher Scientific) and cDNA was made using the Applied Biosystems high-capacity cDNA reverse transcription kit. Primers were obtained from Fisher Scientific and designed for inflammatory markers: Interleukin-1β (IL-1β), Interleukin-1α (IL-1α), Tumor-necrosis factor alpha (TNF-α), Transforming growth factor beta (TGF-β), Lipocalin 2 (LCN2), and chemokine (C-C motif) ligand 2 (CCL2) and lysosomal function gene Cystatin B (CSTB). Changes in relative gene expression were quantified using the $2^{\Delta\Delta CT}$ method and 18S ribosomal RNA as control gene. Details on the primers used can be found in Additional materials and methods section under Supplementary Information.

### Statistics

When evaluating all results, predetermined and appropriate statistical methods are used to determine significance. Cytotoxicity, lysosomal pH and size, western blot protein expression levels, lipid droplets count/cell, serum ALT, BIL and triglyceride levels, liver weights, body weights, and blood glucose levels were expressed as mean ± S.D. Statistical analysis are performed using GraphPad Prism v9, and two-sided unpaired $t$-test was used to determine statistical significance, where $p$-values* $< 0.05$ were considered statistically significant.

### Reporting summary

Further information on research design is available in the Nature Portfolio Reporting Summary linked to this article.

## Data availability

The authors declare that source data are provided with this paper. The PrimerBank IDs for primers used for qPCR analysis can be found in the supplementary information file. Additionally, requests for materials can be also addressed to Professor Mark Grinstaff (mgrin@bu.edu). Source data are provided with this paper.

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

## Acknowledgements

The authors thank those who contributed helpful discussions and sup-port of the research, including Dr. Evan Taddeo, Dr. Jenny Ngo, Dr. Kyle Trudeau, Siyouneh Baghdasarian and Silvana Baghdasarian. We also thank the Boston University Chobanian & Avedisian School of Medi-cine Collaborative Research Laboratory (CoRe) for the processing and interpretation of liver histology sections, Boston University Chobanian & Avedisian School of Medicine Analytical Instrumentation Core for ana-lysis of mice blood and serum samples, Boston University Chobanian & Avedisian School of Medicine Cellular Imaging Core (CIC), and Boston University Micro/Nano Imaging Facility for confocal microscopy equip-ment usage. J.L.Z. and A.M. were supported by a BU Nano Cross-disciplinary fellowship from the BU Nano center at Boston University. J.L.Z. was supported by a Presidential Postdoctoral Fellowship from Nanyang Technological University, Grant/Award Number: 021229-00001. C.H.L. was supported by Dean's Postdoctoral Fellowship, Nanyang Technological University, Lee Kong Chian School of Medicine, Grant/Award Number: 021207-00001. E.A. was supported by Azrieli Fellowship (The Azrieli Foundation). This work was also supported in part by funding from the National Institutes of Health (R01AA026914, OSS ML; R21AG063373, MWG OSS; and R21AG06045, OSS MWG).

## Author contributions

J.L.Z., O.S.S., and M.W.G. conceived the study and designed the experiments. J.L.Z., R.A.P., E.A., A.M., A.J.B., L.F., and C.H.L. carried out data curation and analysis. J.L.Z. conducted the in vitro and in vivo experiments in Boston University (BU), and in Nanyang Technological University (NTU) with assistance from C.H.L. A.M. performed the NP pH titration assay and assisted with glucose and insulin tolerance tests (BU). R.X. assisted with polymer synthesis and characterizations (BU). M.S. assisted J.L.Z. in designing and executing in vivo measurements.

R.A.P. and E.A. performed the mitochondria assays and Rho-acNPs biodistribution data collection and analysis in University of California, Los Angeles (UCLA). L.F. performed insulin signaling activity data collection and analysis (UCLA). A.J.B. performed the RNA expression data collection (UCLA). A.P. assisted with the mice experiments in UCLA. J.L.Z. and MWG wrote the manuscript. R.A.P., E.A., A.M., C.H.L., M.S., M.L., X.H., and O.S.S. edited and provided comments to the manuscript. All authors have given approval to the final version of the manuscript.

## Competing interests

M.W.G., O.S.S., and J.L.Z. are co-inventors on a patent filed and granted in the United States Patent and Trademark Office (Patent number: US10925975B2) on the application of acidic nanoparticles as a treatment for lysosomal acidity compromised diseases. O.S.S. and M.W.G. are co-founders of Enspire Bio/Capacity Bio, which are testing the application of these acidic nanoparticles. The remaining authors declare no other competing interests.

## Ethics and inclusion statement

The experiments and the manuscript writing were performed at four different sites: Boston University (United States of America), Nanyang Technological University (Singapore), University of California, Los Angeles (United States of America), and Shenzhen Middle School (China). We carefully considered researcher contributions and author-ship (as well as inventorship) criteria in this multi-region collaboration to promote greater equity in research collaborations by: (1) ensuring researchers were involved throughout the research process—study design, study implementation, and data ownership; (2) determining inventorship in accordance with patent law, with support and assistance from the technology transfer offices of the universities; (3) assessing the relevancy of the research locality; (4) discussing and agreeing on the roles and responsibilities amongst the collaborators; and (5) determining if the research would be restricted, require local ethics review, local animal welfare review, environmental protection, and/or bio risk-related regulations in the setting of the researcher.
