## [Peer Review File · Nature Communications]

Restoration of Lysosomal Acidification Rescues Autophagy and Metabolic Dysfunction in Non-alcoholic Fatty Liver DiseaseREVIEWER COMMENTS

Reviewer #1 (Remarks to the Author):

The paper reports a nanoparticle that has a novel activity in lowering lysosomal pH and therefore improving autolysosomal homeostasis. When treated to HepG2 cells, this nanoparticle can lower lysosomal pH and increase lysosomal enzymatic activities. It also decreases accumulation of LC3-II and p62, as well as lipid droplets, in palmitate-induced lipotoxic condition. When treated in vivo, it localizes in liver, and improves glucose homeostasis and hepatosteatosis.

Although this is an interesting report, several key experiments need to be conducted to support the main claims.

(1) acNP effects on autophagic flux needs to be shown, in both HepG2 cells and in vivo mouse livers. Currently, only the steady-state levels of LC3 and p62 were shown.

(2) Whether acNP restores in vivo lysosomal homeostasis is not shown. It is unclear whether acNP corrects metabolism through restoring liver metabolism. To address this, hepatocyte lysosomal acidification and lysosomal enzyme activities should be measured in control, LD and HD mice.

(3) Mouse data are predominantly about improvements of insulin-glucose homeostasis, which are not substantiated by in vitro results. Dose acNP restore insulin signaling responsiveness of HepG2 cells?

Reviewer #2 (Remarks to the Author):

NCOMMS-20-24293

Restoration of Lysosomal Acidification Rescues Autophagy and Metabolic Dysfunction in Non-alcoholic Fatty Liver Disease

This study reports an approach to treat non-alcoholic fatty liver disease (NAFLD) using engineered polymeric nanoparticles that can be degraded in lysosomes and recover acidic condition restoring autophagy and metabolic dysfunction. The novelty of this work is the polymer design that allows selective degradation at dysfunctional lysosomal pH and effectively lowers the pH. The authors convincingly demonstrate the potential of this approach in reversing the effects of NAFLD by rescuing autophagic flux in vitro and in vivo. The treatment of NAFLD mice with acNPs led to significantly reduced steatosis, reversal of insulin resistance, as well as decrease in liver weight, and suggests that acNPs has potential as a treatment option for NAFLD.

The manuscript could be strengthened with responses to these major comments:

1. For the general reader of Nature Communications, who might not be familiar with NAFLD and its mechanism, it would be helpful to present a schematic diagram that covers the major pathway addressed here, from lysosomal acidification restored by polymeric nanoparticles and its therapeutic outcomes.
2. The second paragraph of the introduction is almost identical to the introduction of a previous publication by the author with sections that are copied word-for-word (highlighted in manuscript). (Zeng, J.; Shiriha, O. S.; Grinstaff, M. W. Degradable Nanoparticles Restore Lysosomal PH and Autophagic Flux in Lipotoxic Pancreatic Beta Cells. *Adv. Healthc. Mater.* 2019, 8 (12), 1–7.)
3. Curiously, the authors do not show a correlation between polymer degradation, and rate of acidification, which would seem to be the most important structure/function relationship here. A study of nanoparticle degradation (e.g. DLS) and/or polymer degradation (e.g. GPC) should be included to show that pH change correlates with particle degradation. In particular, is the PESU control particle not degrading at all or is the pKa of succinic acid too high to cause any change? As the polymerization of PEFSU is not completely random according to NMR (with sections of homopolymerized succinic acid) maybe the particle/polymer is not completely degrading and only sections of homopolymerized TFSA on the surface are degrading while there are still sections of undegraded TFSA embedded in the nanoparticle core?
4. Why is this particular ratio of TFSA to SA used and not for example a pure homopolymer entirely

composed of TFSA? If other ratios were tested for their pH response, it would be helpful to list those studies in the supplementary information.

5. It would be interesting to see how acNPs affect the pH of normally functioning acidic lysosomes. In particular, would this lead to an excessively low pH?

6. There are many animal models for NAFLD, and the phenotypes are mostly similar but definitely varied. The therapeutic effects of acidic nanoparticles can be also varied, depending on the animal model, nature of induction of disease, and extent of disease progression. Please address the rationale or discussion regarding the specific model used here.

7. The time chart describing animal experiment from disease induction and its treatment should be presented in one of the main figure, since there are groups with different doses and multiple doses.

8. Most nanoparticles—regardless of composition—accumulate in the liver after i.v. administration. However, their distribution within liver tissue are not uniform. In most cases, the majority of particles are taken up by Kupffer cells, not hepatocytes. Considering nanoparticle distribution and their poor diffusion in tissue, it could be unclear how nanoparticles affect lysosome and rescue autophagy. Distribution of nanoparticles using fluorescent microscope imaging should be supported with other types of analysis, such as flow cytometry data.

9. Previous work on nanoparticle-mediated lysosomal acidification should be discussed in more detail. Lysosomal acidification was attempted before in other publications (including by this group) using photoactivated nanoparticles and PLGA nanoparticles with similar studies being conducted.

10. The relationship between lysosome acidification and autophagosome fusion should be discussed. How does lysosome re-acidification rescue autophagosome fusion? If reduced autophagosome fusion is an independent effect of NAFLD how does lysosome acidification affect this?

Other minor comments:

Introduction—Nanoparticles are not always monodisperse and around 100 nm, as suggested here.

Results—What is the pKa of succinic acid, and what role does that play in the action of the particles?

The method used to generate data in Fig 1C is not described in the Methods.

Why don't PESU particles change the pH? Is this because the particles are not degrading (fast enough) or because the pKa of degradation product (succinic acid) is too high to have any effect?

In Fig 5A, how long was treatment to achieve this reduction in liver weight?

Some of the data in Fig 5 could be moved to Supplementary Data to improve readability.

In Fig 6, not all of the changes reached statistical significance, but this is not clear from the text.

Fig 1—Missing subscript on nitrogen: N₂

Reviewer #3 (Remarks to the Author):

Authors investigated a new biodegradable acid-activated acidic nanoparticles (acNPs) as a lysosome targeting strategy to manipulate lysosomal acidity and autophagy in hepatocytes and in experimental non-alcoholic fatty liver disease (NAFLD) mouse model. They were able to show that acNPs, composed of fluorinated polyesters, improved lysosomal acidity and rescued lysosomal function to some extent in cultured human hepatoma cells and in mouse livers. In vivo administration of acNP also can improve diet-induced insulin sensitivity and steatosis likely via increased autophagic flux and mitochondrial functions. While this tool has held a promise to treat lysosomal defects mediated diseases such as NAFLD, the study is largely descriptive in nature. The autophagic flux data and mitophagy were weak and not convincingly support the conclusions.

Specific comments:

1. Figure 2, in addition to the change of lysosome size, it seems that the number of lysosomes also altered. The authors claimed that the size changes could be due to increased turnover via autophagy. However, more data are needed to support this. Authors should quantify the number of lysosomes or more quantitative manner for lysosomal proteins. Lysosomal stress such as pH changes often leads to the activation of TFEB, a master regulator for lysosomal biogenesis gene transcription. Would

altered lysosomal pH affect TFEB-mediated transcription program?

2. Figure 2E, authors should add a lysosomal inhibitor such as Bafilomycin A or leupeptin to confirm the autophagic flux changes of LC3-II. To better support the lipid changes, the total levels of triglyceride should be measured in Figure 2I.

3. Palmitate is toxic to HepG2 cells. Would improved lysosomal functions by acNP affect palmitate lipotoxicity? Ideally these experiments should be repeated in primary cultured hepatocytes as HepG2 cells are cancerous in nature.

4. Figure S3C, the figure labeled as LAMP-1 but in the text it was stated as LAMP-2?

5. More experimental details should be provided for Fig S4.

6. Figure 6, the restoration of autophagy function by using western blot for LC3-II and p62 could be troublesome as both protein could be regulated at the transcription level. Also decreased LC3-II could also be due to decreased formation of autophagosomes. Adding a lysosomal inhibitor such as leupeptin in these experiments will help to clarify these issues. In addition, no data provided to show improved lysosomal functions by acNP in mouse NAFLD models.

7. Inflammation is critical for the progression of NAFLD to NASH. Authors should provide data to show whether acNP can also affect liver non-parenchyma cells (such as macrophages) in addition to hepatocytes. It is highly likely a large amount of acNP would be taken up by macrophages/Kupffer cells in the liver. The function of these macrophage/Kupffer cells after acNP should be determined.

8. Figure 6, mitophagy data were very weak. First the control LFD group was missing, and it was unclear whether HFD would impair mitophagy in this model. The change of mitoTracker could be due to various factors such as mitochondrial membrane potential and may not be a good marker for mitophagy. Mitophagy referred to more specific autophagic removal of damaged mitochondria. Did the authors observe lysosomes that contain mitochondria? If HFD impaired lysosomal pH and functions, one would observe more mitochondria in the lysosomes? And acNP treatment should lead to few mitochondria inside lysosomes. The changes of more mitochondrial proteins should also be included.

Reviewer #1 (Remarks to the Author):

The paper reports a nanoparticle that has a novel activity in lowering lysosomal pH and therefore improving autolysosomal homeostasis. When treated to HepG2 cells, this nanoparticle can lower lysosomal pH and increase lysosomal enzymatic activities. It also decreases accumulation of LC3-II and p62, as well as lipid droplets, in palmitate-induced lipotoxic condition. When treated *in vivo*, it localizes in liver, and improves glucose homeostasis and hepatosteatosis.

Although this is an interesting report, several key experiments need to be conducted to support the main claims.

We thank the reviewer for noting that our work is interesting, and we have performed the suggested additional experiments to substantiate the findings (see below).

(1) acNP effects on autophagic flux needs to be shown, in both HepG2 cells and *in vivo* mouse livers. Currently, only the steady-state levels of LC3 and p62 were shown.

Response: Thank you for the comment and we have performed additional experiments at non steady-state conditions. We have determined the autophagic flux in HepG2 cells via comparing the effects of acNPs addition before and after adding either a lysosomal V-ATPase/acidification inhibitor, bafilomycin A1 (100 nM for 2 hours, Baf), or a lysosomal protease inhibitor, leupeptin (100 uM for 24 hours, Leu) (**Fig. 3E – G**). Bafilomycin disruption of lysosomal acidification resulted in dysfunctional autophagic clearance of autophagosome, hence leading to an accumulation of autophagosomes (e.g., elevation of LC3-II and p62 levels). The addition of leupeptin also resulted in a slight accumulation of LC3-II and p62 levels compared to the acNPs treated condition, although at a lower level than with bafilomycin treatment. These results suggest that the effect of acNPs in modulating autophagic flux is mediated mainly through affecting lysosomal acidification, and not due to modulating the lysosomal protease degradative activity. We have henceforth also chosen bafilomycin as the standard control in the subsequent experiments.

The study of autophagic flux in mouse livers using lysosomal acidification inhibitors (e.g., chloroquine, bafilomycin) is limited by the toxicity of these treatments (e.g., cardiac, neurotoxicity and retinal toxicity and cell death)¹. Primary human hepatocytes are considered the gold standard short-term human *in vitro* liver model because of their high functionality relative to the human organ *in vivo*². Hence, we chose to use primary human hepatocytes and analyzed the turnover of mitochondria (e.g., mitophagy) to determine autophagic flux *in vivo*. Using primary human hepatocytes, we show that under palmitate conditions (recapitulate HFD condition in mice), mitochondrial content is increased while treatment with acNPs decreases it, indicating increased mitochondrial degradation (**Fig. S7K**). The maximal respiratory rate

(MRR) of mitochondria decreases under palmitate condition, while the addition of acNPs restores the MRR (**Fig. S7L**). When we measured the oxygen consumption rate (OCR) of mitochondria with and without lysosomal acidification with bafilomycin or acNPs, the OCR of mitochondria decreases upon bafilomycin treatment, while acNPs treatment restores OCR (**Fig. S7M**). In sum, these data indicate that the effect of acNPs in restoring mitochondrial respiratory function is due to improving lysosomal acidification function and autophagic turnover of mitochondria.

(2) Whether acNP restores *in vivo* lysosomal homeostasis is not shown. It is unclear whether acNP corrects metabolism through restoring liver metabolism. To address this, hepatocyte lysosomal acidification and lysosomal enzyme activities should be measured in control, LD and HD mice.

Response: Currently, there are no tools to directly measure lysosomal acidification and lysosomal enzyme activities *in vivo*. Therefore, we acknowledge this is a limitation in the study. To pinpoint the effect of lysosomal acidification on lysosomal homeostasis *in vivo*, we have treated primary human hepatocytes in the presence of bafilomycin or palmitate, with or without acNPs, and measured the lysosomal activity using Magic red assay (fluorescence increases intensity as lysosomal cathepsin L activity increases) (**Fig. S7N – O**). Treatment with either Bafilomycin A1 or palmitate decreases Magic red intensity, indicating decreased lysosomal enzyme activity. Treatment with low dose (LD) or high dose (HD) acNPs increases the magic red intensity, and HD acNPs show a more significant increase. These results indicate that the increase in lysosomal enzyme activity is due to changes in lysosomal acidification upon acNPs treatment.

(3) Mouse data are predominantly about improvements of insulin-glucose homeostasis, which are not substantiated by *in vitro* results. Dose acNP restore insulin signaling responsiveness of HepG2 cells?

Response: To determine if acNPs restore insulin responsiveness of HepG2 cells, we determined insulin signaling responsiveness through blotting for insulin signaling pathway proteins, after stimulation of HepG2 cells treated with BSA, palmitate, or palmitate and acNPs, with 100 nM insulin for 15 minutes. In the insulin signaling pathway, insulin binds to insulin receptor (IRS1), and IRS1 is phosphorylated to initiate downstream signaling of PI3K. Akt is activated in response to PI3K signaling, and the activated Akt phosphorylates GSK3 β at serine 9, leading to GSK3 β inhibition. In HepG2 cells under palmitate treatment, there is reduced phosphorylation of IR due to increased hyper-phosphorylation of IRS-1. The inhibition of Akt (i.e., reduced Akt phosphorylation) activates GSK3 β through reducing GSK3 β phosphorylation at serine 9. Treatment with acNPs increases p-Akt and subsequently p-GSK3 β levels,

thereby indicating an increase in insulin response in HepG2 cells. These results are found in **Figure. S2H – K**.

Reviewer #2 (Remarks to the Author):

NCOMMS-20-24293

Restoration of Lysosomal Acidification Rescues Autophagy and Metabolic Dysfunction in Non-alcoholic Fatty Liver Disease

This study reports an approach to treat non-alcoholic fatty liver disease (NAFLD) using engineered polymeric nanoparticles that can be degraded in lysosomes and recover acidic condition restoring autophagy and metabolic dysfunction. The novelty of this work is the polymer design that allows selective degradation at dysfunctional lysosomal pH and effectively lowers the pH. The authors convincingly demonstrate the potential of this approach in reversing the effects of NAFLD by rescuing autophagic flux in vitro and in vivo. The treatment of NAFLD mice with acNPs led to significantly reduced steatosis, reversal of insulin resistance, as well as decrease in liver weight, and suggests that acNPs has potential as a treatment option for NAFLD.

We thank the reviewer for noting the novelty of our work.

The manuscript could be strengthened with responses to these major comments:

1. For the general reader of Nature Communications, who might not be familiar with NAFLD and its mechanism, it would be helpful to present a schematic diagram that covers the major pathway addressed here, from lysosomal acidification restored by polymeric nanoparticles and its therapeutic outcomes.
Response: Thank you for the comment. A schematic which links lysosomal acidification to autophagy, mitochondrial function, lipid accumulation, and inflammation in the pathogenesis of NAFLD, as well as how NPs treatment affects these processes, has been included in **Figure. 1** of the revised manuscript.

2. The second paragraph of the introduction is almost identical to the introduction of a previous publication by the author with sections that are copied word-for-word (highlighted in manuscript). (Zeng, J.; Shirihai, O. S.; Grinstaff, M. W. Degradable Nanoparticles Restore Lysosomal PH and Autophagic Flux in Lipotoxic Pancreatic Beta Cells. Adv. Healthc. Mater. 2019, 8 (12), 1–7.)

Response: We have rewritten the second paragraph, and this is reflected in the main text of the revised manuscript.

3. Curiously, the authors do not show a correlation between polymer degradation, and rate of

acidification, which would seem to be the most important structure/function relationship here. A study of nanoparticle degradation (e.g. DLS) and/or polymer degradation (e.g. GPC) should be included to show that pH change correlates with particle degradation. In particular, is the PESU control particle not degrading at all or is the pKa of succinic acid too high to cause any change? As the polymerization of PEFSU is not completely random according to NMR (with sections of homopolymerized succinic acid) maybe the particle/polymer is not completely degrading and only sections of homopolymerized TFSA on the surface are degrading while there are still sections of undegraded TFSA embedded in the nanoparticle core?

Response: We have included a GPC study to analyze the molecular weight of the polymers after degradation in pH 6.0 buffer. We observed a correlation between polymer degradation and buffer acidification. The 25% PEFSU polymer degrades rapidly in the first 24 hours, followed by a much slower rate of degradation within the next 24 hours (**Fig. 2E**). We have also used scanning electron microscopy to monitor the size changes of the nanoparticles across 48 hours, where the nanoparticles size increases within the first 24 hours, potentially indicative of bulk degradation (**Fig. S1H – J**). This aligns with other studies describing polyester (e.g., PLGA) nanoparticle degradation with an initial particle swelling before a decrease in nanoparticle size³.

The pKas of succinic acid are 4.21 and 5.64. The PESU control NPs have a very slow rate of degradation over 48 hours. It is likely due to the PESU being a more crystalline polymer ($T_m = 111^\circ\text{C}$), while 25% PEFSU is an amorphous polymer ($T_g = -27^\circ\text{C}$), resulting in a tightly embedded polymer in the nanoparticle core, resulting in its low rate of degradation. Hence, within the time frame of our assay, the PESU control polymer does not result in significant changes in the lysosomal pH.

4. Why is this particular ratio of TFSA to SA used and not for example a pure homopolymer entirely composed of TFSA? If other ratios were tested for their pH response, it would be helpful to list those studies in the supplementary information.

Response: We thank the reviewer for the suggestion. We have previously tried to make a pure homopolymer entirely composed of TFSA, however the polymerization was unsuccessful despite optimization using different reaction conditions and parameters. We have included a table of polymers that we have successfully synthesized with different ratios of TFSA and SA, as well as with different glycol linker groups (**Fig. A**). We have also determined the pH modulation capability of these polymers, and it was shown that 25% TFSA to 75% SA (25% PEFSU) had the most significant pH modulation property (**Fig. B**). We will discuss the studies briefly in the manuscript text, but we do not intend to

include these data in the supplementary information, as these results will be discussed in greater detail in subsequent publications focused mainly on the polymer chemistry.

Polymer name	Yield (%)	X:Y (Feed)	X:Y (NMR)	Glycol	\bar{M}_n (g/mol)	M_w (g/mol)	\bar{D}
10% PEFSU	85.0	10:90	10:90	Ethylene	5764	7205	1.25
15% PEFSU	85.0	15:85	14:86	Ethylene	6881	7293	1.06
20% PEFSU	84.0	20:80	21:79	Ethylene	6862	7891	1.15
25% PEFSU	78.0	25:75	24:86	Ethylene	10082	12098	1.20
PESU	92.0	0:100	0:100	Ethylene	9740	11104	1.14
25% PPFSU	70.0	25:75	27:73	Propylene	16526	23896	1.47
50% PPFSU	72.0	50:50	53:47	Propylene	10489	13425	1.28
75% PPFSU	73.0	75:25	72:27	Propylene	15740	23610	1.50
100% PPFSU	85.0	100:0	100:0	Propylene	12402	15750	1.27
PPSU	92.0	0:100	0:100	Propylene	11479	15956	1.39
25% PBFSU	87.0	25:75	25:75	Butylene	9077	13615	1.50
50% PBFSU	93.0	50:50	52:48	Butylene	14502	22478	1.55
75% PBFSU	84.0	75:25	77:23	Butylene	11587	19299	1.66
100% PBFSU	94.0	100:0	100:0	Butylene	14154	18117	1.28
PBSU	95.0	0:100	0:100	Butylene	11945	15289	1.28

5. It would be interesting to see how acNPs affect the pH of normally functioning acidic lysosomes. In particular, would this lead to an excessively low pH?

Response: We have measured the lysosomal pH in HepG2 cells with acNPs treatment in BSA media and the pH does not change significantly with respect to the BSA control cells (**Fig. S2A – B**).

6. There are many animal models for NAFLD, and the phenotypes are mostly similar but definitely varied. The therapeutic effects of acidic nanoparticles can be also varied, depending on the animal model, nature of induction of disease, and extent of disease progression. Please address the rationale or discussion regarding the specific model used here.

Response: Mice and rats have been used most frequently in NAFLD modeling. The C57BL/6 strain in mice and Wistar and Sprague Dawley strains in rats are generally preferred because of their intrinsic preference

to develop obesity and NAFLD^{4,5}. Hence, we have used C57BL/6 strain mice as our high-fat diet (HFD) model. There have been multiple diets used to generate NAFLD phenotypes, including MCD diet, Choline-Deficient L-Amino Acid-defined Diet, Atherogenic Diet, Fructose and High-fat diet (HFD). We chose the HFD fed model with fat content of 60 kcal%, as it has been shown to bring about a phenotype similar to the human disease, characterized by obesity (after 10 weeks), insulin resistance (hyperinsulinemia after 10 weeks and glucose intolerance after 12 weeks) and hyperlipidemia (after 10 weeks)^{6,7}. 16 weeks of HFD regimen was chosen as it recapitulates liver steatosis found in NAFLD, and evidence suggests that mice fed with HFD for 16 weeks had decreased mitochondrial respiration and bioenergetics compared with control mice⁸. We have included this additional information on animal model choice in the revised manuscript.

7. The time chart describing animal experiment from disease induction and its treatment should be presented in one of the main figure, since there are groups with different doses and multiple doses.

Response: We thank the reviewer for the comment and have included a time chart describing the animal experiment and different dosage regimen in the revised manuscript (**Fig. 4A**).

8. Most nanoparticles—regardless of composition—accumulate in the liver after i.v. administration. However, their distribution within liver tissue are not uniform. In most cases, the majority of particles are taken up by Kupffer cells, not hepatocytes. Considering nanoparticle distribution and their poor diffusion in tissue, it could be unclear how nanoparticles affect lysosome and rescue autophagy. Distribution of nanoparticles using fluorescent microscope imaging should be supported with other types of analysis, such as flow cytometry data.

Response: We studied the co-localization of Rhodamine labelled acNPs in HFD mice with Kupffer cells stained with alexa-fluor-647-anti-mouse-clec4f-antibody (BioLegend) upon tail vein injection. Through quantification of the immunofluorescence images, we observed that only 20% of the Rho-acNPs co-localized with MAC2 positive cells, while 80% of the Rho-acNPs resided in hepatocytes or the surrounding tissues. Hence, this indicates that the effect of acNPs on autophagic function as well as liver metabolic function is largely due to its action in hepatocytes. These data are included in **Figure. S3C**.

9. Previous work on nanoparticle-mediated lysosomal acidification should be discussed in more detail. Lysosomal acidification was attempted before in other publications (including by this group) using photoactivated nanoparticles and PLGA nanoparticles with similar studies being conducted.

Response: We thank the reviewer for the suggestion. We have included the discussion of previous work on nanoparticle-mediated lysosomal acidification using the photoactivated nanoparticles and PLGA nanoparticles in the revised manuscript. While the photoactivated NPs show significant effect in restoring

lysosomal acidity in pancreatic beta cells under lipotoxicity (type II diabetes model)⁹, the requirement of a UV-light trigger to activate the nanoparticles renders it inapplicable for *in vivo* applications. PLGA NP has been used to modulate lysosomal pH in a variety of disease models^{10,11}. The acNP exhibit a 4 times more significant lysosomal pH reduction in HepG2 cells under palmitate treatment than PLGA NP at the same concentration (**Fig. S8A**). In addition, acNP causes a more significant reduction in lipid droplets accumulation in HepG2 cells, potentially due to a more significant decrease in lysosomal pH (**Fig. S8B**).

10. The relationship between lysosome acidification and autophagosome fusion should be discussed. How does lysosome re-acidification rescue autophagosome fusion? If reduced autophagosome fusion is an independent effect of NAFLD how does lysosome acidification affect this?

In NAFLD, increases in intracellular lipids (e.g., saturated fatty acids such as palmitic acid) have been shown to alter the intracellular membrane lipid composition of both autophagosomes and lysosomes. Hence, this reduces the ability of autophagosomes to fuse with lysosomes and a subsequent decrease in autophagic flux¹². Lysosomal pH has been shown to play a role in modulating autophagosome-lysosome fusion. In our previous work using photo-activated nanoparticles (paNPs) to re-acidify lysosomes in pancreatic β cells under lipotoxicity (palmitate treatment) (Assali et al. (2019) *FASEB* 33(3), pp 4154-4165)¹³, we addressed the effect of lysosomal reacidification on mitochondrial turnover (mitophagy) using a mCherry-GFP-Fis1 probe, a mitochondrial chimeric protein that allows the identification of colocalization of mitochondria (in autophagosomes) inside autolysosomes. At neutral pH, the mitochondria emit both red and green fluorescence. Upon fusion with the acidic lysosomes, the low pH quenches the GFP signal, without affecting mCherry fluorescence. Under palmitate treatment, the number of red mitochondria in lysosomes is decreased, indicating a reduction of mitophagy. Upon lysosome re-acidification with paNPs, the number of red mitochondria (in autophagosomes) increases, indicating a restoration of mitophagy as a result of increased autophagosome-lysosome fusion¹³.

Other minor comments:

Introduction—Nanoparticles are not always monodisperse and around 100 nm, as suggested here.

Response: We agree and changed the corresponding text.

Results—What is the pKa of succinic acid, and what role does that play in the action of the particles?

The method used to generate data in Fig 1C is not described in the Methods. Why don't PESU particles change the pH? Is this because the particles are not degrading (fast enough) or because the pKa of degradation product (succinic acid) is too high to have any effect?

Response: The pKas of succinic acid are 4.21 and 5.64. The polymer formed from pure SA has a high Tm as compared to 25% PEFSU, hence, within the time frame of our assay, the PESU polymer does not have significant degradation and does not generate significant changes in the lysosomal pH. We have included the method to generate data in **Fig. 2C** (previously Fig. 1C) in the materials and methods section.

In Fig 5A, how long was treatment to achieve this reduction in liver weight?

Response: The treatment was for three times (e.g., day 1, day 3 and day 5). We have also added this detail in the newly included timeline (**Fig. 4A**) for this experiment in the revised manuscript.

Some of the data in Fig 5 could be moved to Supplementary Data to improve readability.

Response: We have moved the liver weight/body weight % data for both HFD and LFD conditions in Fig. 5 to the supplementary data to improve readability.

In Fig 6, not all of the changes reached statistical significance, but this is not clear from the text.

Response: We have changed our text to describe these changes, as well as discussion of the results.

Fig 1—Missing subscript on nitrogen: N2

Response: We have changed the subscript on nitrogen to be N₂.

Reviewer #3 (Remarks to the Author):

Authors investigated a new biodegradable acid-activated acidic nanoparticles (acNPs) as a lysosome targeting strategy to manipulate lysosomal acidity and autophagy in hepatocytes and in experimental non-alcoholic fatty liver disease (NAFLD) mouse model. They were able to show that acNPs, composed of fluorinated polyesters, improved lysosomal acidity and rescued lysosomal function to some extent in cultured human hepatoma cells and in mouse livers. In vivo administration of acNP also can improve diet-induced insulin sensitivity and steatosis likely via increased autophagic flux and mitochondrial functions. While this tool has held a promise to treat lysosomal defects mediated diseases such as NAFLD, the study is largely descriptive in nature. The autophagic flux data and mitophagy were weak and not convincingly support the conclusions.

We thank the reviewer for the critical comments and we have performed additional experiments to support our findings.

Specific comments:

1. Figure 2, in addition to the change of lysosome size, it seems that the number of lysosomes also altered. The authors claimed that the size changes could be due to increased turnover via autophagy. However, more data are needed to support this. Authors should quantify the number of lysosomes or more quantitative manner for lysosomal proteins. Lysosomal stress such as pH changes often leads to the activation of TFEB, a master regulator for lysosomal biogenesis gene transcription. Would altered lysosomal pH affect TFEB-mediated transcription program?

Response: We have quantified the number of lysosomes for each treatment condition. There is a decrease in the average number of lysosomes under palmitate treatment, however; the changes observed across treatment conditions are not statistically significant (see **Figure. S2C**). In a similar study done by our group studying the change in lysosomal acidification and lysosome number in pancreatic β cells under palmitate treatment, we showed that there is no change in the total lysosomal mass/number before and after palmitate treatment, as determined by LAMP-1 staining, but there is lysosomal pH increase after palmitate treatment, indicative that the change in lysosomal acidification does not affect lysosome number¹³.

2. Figure 2E, authors should add a lysosomal inhibitor such as Bafilomycin A or leupeptin to confirm the autophagic flux changes of LC3-II. To better support the lipid changes, the total levels of triglyceride should be measured in Figure 2I.

Response: Thank you for the comment. We have determined the autophagic flux in HepG2 cells via comparing the effects of acNPs addition before and after adding either a lysosomal V-ATPase/acidification inhibitor, bafilomycin A1 (100 nM for 2 hours, Baf), or a lysosomal protease inhibitor, leupeptin (100 μ M for 24 hours, Leu) (**Fig. 2E – G**). Bafilomycin disruption of lysosomal acidification results in dysfunctional autophagic clearance of autophagosome, hence leading to an accumulation of autophagosomes (e.g., elevation of LC3-II and p62 levels). The addition of leupeptin also results in a slight accumulation of LC3-II and p62 levels compared to the acNPs treated condition, although at a lower level than with bafilomycin treatment. These results suggest that the effect of acNPs in modulating autophagic flux is mediated mainly through affecting lysosomal acidification, and not due to modulating the lysosomal protease degradative activity. We have also measured the total levels of triglyceride in HepG2 cells (**Fig. S2E**).

3. Palmitate is toxic to HepG2 cells. Would improved lysosomal functions by acNP affect palmitate lipotoxicity? Ideally these experiments should be repeated in primary cultured hepatocytes as HepG2 cells are cancerous in nature.

Response: We conducted cell viability assays in both A) HepG2 cells, and B) primary human hepatocytes. Addition of palmitate to HepG2 or primary human hepatocytes result in a 20 – 25% reduction in cell viability, and treatment with acNPs (100 µg/mL) restores this cell viability (**Figure. S2F**) However, in the primary human hepatocytes, higher doses of acNPs (> 120 µg/mL), results in additional cell cytotoxicity (**Figure. S2G**).

4. Figure S3C, the figure labeled as LAMP-1 but in the text it was stated as LAMP-2?

Response: We have changed the text to LAMP-1 to ensure consistency.

5. More experimental details should be provided for Fig S4.

Response: We have included more experimental details for Fig. S4 in the revised supplementary information document, under the section “Blood and serum analysis”.

6. Figure 6, the restoration of autophagy function by using western blot for LC3-II and p62 could be troublesome as both protein could be regulated at the transcription level. Also decreased LC3-II could also be due to decreased formation of autophagosomes. Adding a lysosomal inhibitor such as leupeptin in these experiments will help to clarify these issues. In addition, no data provided to show improved lysosomal functions by acNP in mouse NAFLD models.

Response: In the HepG2 cells, the addition of leupeptin does not result in a significant accumulation of LC3-II and p62 (**Fig. 3E – G**), but addition of bafilomycin results in a significant accumulation of LC3-II and p62. This indicates that the effect of acNPs in modulating autophagic flux is mediated mainly through changes in lysosomal acidification, and not due to modulating the lysosomal protease degradative activity. Therefore, we propose addition of a lysosomal acidification inhibition control will be more suitable to clarify the issues, as mentioned by the reviewer.

Using primary human hepatocytes, we show that under palmitate conditions (recapitulate HFD condition in mice), mitochondrial content is increased, while addition of acNPs decreases it, indicating increased mitochondrial degradation (**Fig. S7K**). The maximal respiratory rate (MRR) of mitochondria decrease under palmitate condition, while the addition of acNPs restores the MRR (**Fig. S7L**). When we measured the oxygen consumption rate (OCR) of mitochondria with and without lysosomal acidification with bafilomycin or acNPs, the OCR of mitochondria decreases upon bafilomycin treatment, while acNPs addition restores OCR (**Fig. S7M**). In sum, these data indicate that the effect of acNPs in restoring mitochondrial respiratory function is due to improving lysosomal acidification function, and autophagic turnover of mitochondria.

(2) Whether acNP restores *in vivo* lysosomal homeostasis is not shown. It is unclear whether acNP corrects metabolism through restoring liver metabolism. To address this, hepatocyte lysosomal acidification and lysosomal enzyme activities should be measured in control, LD and HD mice.

Response: Currently, there are no tools to directly measure lysosomal acidification and lysosomal enzyme activities *in vivo*. Therefore, we acknowledge this is a limitation in the study. To pinpoint the effect of lysosomal acidification on lysosomal homeostasis *in vivo*, we have treated primary human hepatocytes in the presence of bafilomycin and acNPs and measured the lysosomal activity using Magic red assay (fluorescence increases intensity as lysosomal cathepsin L activity increases) (**Fig. S7N – O**). Treatment with either Bafilomycin A1 or palmitate decreases Magic red intensity, indicating decreased lysosomal enzyme activity. Treatment with LD or HD acNPs increases the magic red intensity, with HD acNPs showing a more significant increase, and hence increase in lysosomal enzyme activity. These results indicate that the increase in lysosomal activity, due to treatment with acNPs, is due to changes in lysosomal acidification.

7. Inflammation is critical for the progression of NAFLD to NASH. Authors should provide data to show whether acNP can also affect liver non-parenchyma cells (such as macrophages) in addition to hepatocytes. It is highly likely a large amount of acNP would be taken up by macrophages/Kupffer cells in the liver. The function of these macrophage/Kupffer cells after acNP should be determined.

Response: We studied the co-localization of Rhodamine labelled acNPs in HFD mice with Kupffer cells stained with alexa-fluor-647-anti-mouse-clec4f-antibody (BioLegend)¹⁴ upon tail vein injection. Through quantification of the immunofluorescence images, we observed that only 20% of the Rho-acNPs co-localized with MAC2 positive cells, while 80% of the Rho-acNPs resided in hepatocytes or the surrounding tissues. Hence, this indicates that the effect of acNPs on autophagic function as well as liver metabolic function is largely due to its action in hepatocytes. Please see the new data and results in **Figure S3C**.

8. Figure 6, mitophagy data were very weak. First the control LFD group was missing, and it was unclear whether HFD would impair mitophagy in this model. The change of mitoTracker could be due to various factors such as mitochondrial membrane potential and may not be a good marker for mitophagy.

Mitophagy referred to more specific autophagic removal of damaged mitochondria. Did the authors observed lysosomes that contain mitochondria? If HFD impaired lysosomal pH and functions, one would observe more mitochondria in the lysosomes? And acNP treatment should lead to few mitochondria inside lysosomes. The changes of more mitochondrial proteins should also be included.

Response: We thank the reviewer for the comments. We have included the LFD control, LFD with LD acNPs and LFD with HD acNPs mitochondria respirometry data (**Fig. S7H – I**), as well as the mitochondrial content determined using MitoTracker Deep Red (**Fig. 7G – H**).

We did the mitochondrial respirometry on fresh liver lysates for the HFD control, HFD and LD acNPs and HFD and HD acNPs. However, we only have frozen liver lysates for the LFD control, LFD and LD acNPs and LFD and HD acNPs. In fresh liver lysates, high dose acNPs shows an increase in MRR on pyruvate+malate (equivalent of NADH in frozen) and a slight increase in MRR on succinate. The increase in MRR that was seen in fresh lysates can be reproduced in frozen liver lysates when only maximal respiratory rate was measured. Therefore, we only measured MRR in the LFD samples. The addition of LD or HD acNPs did not change the Complex I or IV MRR in LFD mice. acNPs treatment slightly increases the maximal respiratory activity of complex II, although not statistically significant. In frozen liver lysates of HFD mice group, maximal respiratory rates of Complex I and II are less than that of LFD mice. These data show that acNPs addition does not affect mitochondrial function in LFD mice (**Fig. S7H – I**), and are included in the manuscript.

The MTDR dye was chosen because it has been previously used as a dye to determine mitochondrial content independently of their membrane potential and their ultrastructure. In a recently published article done by our group (Acin-Perez et al. (2020), *EMBO J* (2020)39:e104073)¹⁵, we compared the ability of two mitochondrial dyes – MitoTracker Red (MTR), and MitoTracker Deep Red (MTDR, as used in this study), on pancreatic-derived INS1 cells after fixation with paraformaldehyde (PFA), and fixation and permeabilization with PFA and Triton X100. MTDR staining was preserved while MTR diffused after fixation and permeabilization. When freshly isolated mitochondria were stained with MTR or MTDR before and after depolarizing them with either FCCP or calcium overload, membrane potential disruption was only observed in MTR staining and not MTDR staining. Hence, we have shown the MTDR is a viable marker for mitochondrial protein content. The treatment of the LD or HD groups with acNPs increases the activity of both Complex I and II without affecting Complex IV. In LFD group, the mitochondrial content did not increase across all experimental groups, while the mitochondrial content of HFD group increased, and addition of acNPs decreased the mitochondrial content (**Fig. 7G – H**).

We have previously investigated the effect of photo-activated NPs (i.e., a nanoparticle that release carboxylic acid to modulate lysosomal pH upon application of UV-light) on mitophagy in pancreatic beta cells under lipotoxicity (Assali et al. (2019) *FASEB* 33(3), pp 4154-4165)¹³. We studied the effect of these nanoparticles on lysosomal reacidification on mitophagy using a mCherry-GFP-Fis1 probe, a

mitochondrial chimeric protein that allows the identification of mitochondria inside autolysosomes. Under neutral pH conditions, mitochondria emit red and green fluorescence (yellow mitochondria as shown in 'BSA condition'). When mitochondria are recruited into autolysosomes, mitochondria are exposed to low pH that quenches the green fluorescent protein signal without affecting mCherry fluorescence. As shown in Fig. 3E, F from the paper below, palmitate treatment reduced the number of red mitochondria indicating a reduction of mitophagy. This effect was reversed by treatment with nanoparticles that re-acidified the lysosomes. Hence, these results demonstrate the capacity of acidifying nanoparticles to restore mitophagy.

References:

1. Gaynes, B. I., Torczynski, E., Varro, Z., Grostern, R. & Perlman, J. Retinal toxicity of chloroquine hydrochloride administered by intraperitoneal injection. *J. Appl. Toxicol.* **28**, 895–900 (2008).
2. Müller, F. A. & Sturla, S. J. Human in vitro models of nonalcoholic fatty liver disease. *Curr. Opin. Toxicol.* **16**, 9–16 (2019).
3. Zhang, H., Cui, W., Bei, J. & Wang, S. Preparation of poly(lactide-co-glycolide-co-caprolactone) nanoparticles and their degradation behaviour in aqueous solution. *Polym. Degrad. Stab.* **91**, 1929–1936 (2006).
4. Kohli, R. & Feldstein, A. E. NASH animal models: are we there yet? *J. Hepatol.* **55**, 941–943 (2011).
5. Takahashi, Y., Soejima, Y. & Fukusato, T. Animal models of nonalcoholic fatty liver disease/nonalcoholic steatohepatitis. *World journal of gastroenterology* vol. 18 2300–2308 (2012).
6. Van Herck, M. A., Vonghia, L. & Francque, S. M. Animal Models of Nonalcoholic Fatty Liver Disease-A Starter's Guide. *Nutrients* **9**, 1072 (2017).

7. Jahn, D., Kircher, S., Hermanns, H. M. & Geier, A. Animal models of NAFLD from a hepatologist's point of view. *Biochim. Biophys. Acta - Mol. Basis Dis.* **1865**, 943–953 (2019).
8. Eccleston, H. B. *et al.* Chronic exposure to a high-fat diet induces hepatic steatosis, impairs nitric oxide bioavailability, and modifies the mitochondrial proteome in mice. *Antioxid. Redox Signal.* **15**, 447–459 (2011).
9. Trudeau, K. M. *et al.* Lysosome acidification by photoactivated nanoparticles restores autophagy under lipotoxicity. *J. Cell Biol.* **214**, 25–34 (2016).
10. Zeng, J., Martin, A., Han, X., Shirihai, O. S. & Grinstaff, M. W. Biodegradable PLGA Nanoparticles Restore Lysosomal Acidity and Protect Neural PC-12 Cells against Mitochondrial Toxicity. *Ind. Eng. Chem. Res.* **58**, 13910–13917 (2019).
11. Zeng, J., Shirihai, O. S. & Grinstaff, M. W. Degradable Nanoparticles Restore Lysosomal pH and Autophagic Flux in Lipotoxic Pancreatic Beta Cells. *Adv. Healthc. Mater.* 1801511 (2019) doi:10.1002/adhm.201801511.
12. Koga, H., Kaushik, S. & Cuervo, A. M. Altered lipid content inhibits autophagic vesicular fusion. *FASEB J. Off. Publ. Fed. Am. Soc. Exp. Biol.* **24**, 3052–3065 (2010).
13. Assali, E. A. *et al.* Nanoparticle-mediated lysosomal reacidification restores mitochondrial turnover and function in β cells under lipotoxicity. *FASEB J.* **33**, 4154–4165 (2019).
14. Hsieh, S.-L. E. & Yang, C.-Y. CLEC4F, A Kupffer Cells Specific Marker, Is Critical for Presentation of Alfa-Galactoceromide to NKT Cells (78.38). *J. Immunol.* **182**, 78.38 LP-78.38 (2009).
15. Acin-Perez, R. *et al.* A novel approach to measure mitochondrial respiration in frozen biological samples. *EMBO J.* **39**, e104073 (2020).

REVIEWER COMMENTS

Reviewer #1 (Remarks to the Author):

The manuscript was improved by partially addressing the points raised. However, the authors did not completely address the issue since the experiments are missing critical controls. Some issues are reiterated here:

(1) Autophagy flux was only measured in AcNP-treated cells (Fig. 3E). This needs to be examined in controls, such as untreated and PA-treated cells. The current experiments show that there are some flux after AcNP treatments. However, it does not show whether it is an improvement or not.

(2) Insulin sensitivity, not the basal insulin signaling activity needs to be measured (Fig. S2H). The cells need to be treated with insulin, and the difference from the basal level is the important output. The current experiments only show the basal level, not the insulin-stimulated level.

Reviewer #2 (Remarks to the Author):

The authors have responded to all of our previous critiques in a helpful and constructive way. The revised submission is greatly improved as a result of thorough responses to all of the reviewer comments.

Reviewer #3 (Remarks to the Author):

This is a revised manuscript. Authors have performed additional experiments and most previous concerns have been addressed. However, this reviewer still has some issues.
Specific comments:

The conclusion that acNPs increased mitophagy/ mitochondrial degradation (Fig. S7K) was weak, as it was mainly based on the Fig S7E that appeared only CV had significant changes. More mitochondrial proteins should be assessed to strengthen this conclusion.

In many western blots such as Figure 7A, GAPDH was used as loading controls but there were large variations among the samples? Any explanation? It was unclear for the conclusions drawn based on these blots for densitometry analysis? Perhaps another loading control such as beta-actin should be used to validate these results and conclusions.

Reviewer comments:

Reviewer #1 (Remarks to the Author):

Response: The manuscript was improved by partially addressing the points raised. However, the authors did not completely address the issue since the experiments are missing critical controls. Some issues are re-iterated here:

Comment: Thank you for re-reading the manuscript and providing additional critical comments.

Comment 1: Autophagy flux was only measured in AcNP-treated cells (Fig. 3E). This needs to be examined in controls, such as untreated and PA-treated cells. The current experiments show that there are some flux after AcNP treatments. However, it does not show whether it is an improvement or not.

Response 1: We thank the reviewer for this important point. We have now measured the autophagic flux in all treatment conditions, including untreated control, palmitate-only treated cells, palmitate treated cells with acNP. The results indicate that there is an improvement in the autophagic flux upon acNP treatment (2-fold changes) compared to palmitate-only treated cells. These data are now illustrated in Fig. 3E – G of the revised manuscript. Please see page 6.

Comment 2: Insulin sensitivity, not the basal insulin signaling activity needs to be measured (Fig. S2H). The cells need to be treated with insulin, and the difference from the basal level is the important output. The current experiments only show the basal level, not the insulin-stimulated level.

Response 2: Thank you very much for the comment and the suggestion. We have now measured the insulin sensitivity by taking into account both the basal insulin signaling activity and the insulin-stimulated level. The result shows that acNPs increase insulin sensitivity. These data are now illustrated in Fig. S2H – K of the revised manuscript. Please see page 6.

Reviewer #2 (Remarks to the Author):

Comment 1: The authors have responded to all of our previous critiques in a helpful and constructive way. The revised submission is greatly improved as a result of thorough responses to all of the reviewer comments.

Response 1: Thank you very much for your useful comments and critiques in helping us improve the manuscript.

Reviewer #3 (Remarks to the Author):

Comment: This is a revised manuscript. Authors have performed additional experiments and most previous concerns have been addressed. However, this reviewer still has some issues.

Response: Thank you. We appreciated your critical comments on the original manuscript and those suggestions improved the manuscript. Again, we thank you for taking the time to read the manuscript and the new comments below.

Specific comments:

Comment 1: The conclusion that acNPs increased mitophagy/ mitochondrial degradation (Fig. S7K) was weak, as it was mainly based on the Fig S7E that appeared only CV had significant

changes. More mitochondrial proteins should be assessed to strengthen this conclusion.

Response 1: We thank the reviewer for this critique. Our primary goal is to study whether an improvement in lysosomal acidification leads to an end-point result in improvement in mitochondrial function. An initial observation, both in the mouse liver lysates and primary human hepatocytes, shows that there is no significant changes in mitochondria content upon acNPs treatment, suggesting that there is no change in mitophagy, which was not a main focus of our paper. Since there is no change in mitochondria content in both models, and we have indicated that there is no change in mitophagy, we did not blot for more mitochondrial proteins. Importantly, there is a consistent observation in an improvement in mitochondria oxidative respiratory consumption rate, indicative of improved mitochondria functional quality arising from lysosomal re-acidification, with treatment of acNPs. We have removed the brief reference to mitophagy in the revised manuscript and edited the text to be clearer. Future studies will investigate mitophagy and its potential role. Please see Figure 7 and S7.

Comment 2: In many western blots such as Figure 7A, GAPDH was used as loading controls but there were large variations among the samples? Any explanation? It was unclear for the conclusions drawn based on these blots for densitometry analysis? Perhaps another loading control such as beta-actin should be used to validate these results and conclusions.

Response 2: Thank you for pointing this out. Loading controls for Western blots were illustrated in Fig. 3E, Fig S2H, Fig 7A and Fig 7D. The loading controls in Fig. 3E and Fig. S2H, which are cell-based data, are relatively consistent between the treatment conditions. We have re-probed for beta actin in Fig. 7A and Fig. 7D (mouse tissue), and are now included in the revised manuscript. Both GAPDH and beta-actin loading controls show similar trends, indicating that the fluctuation is due to loading variations rather than biological effects. This is supported by several other studies, including ours, showing that lysosomal re-acidification does not affect loading controls.¹⁻³ Please see Figure 7.

References

1. Zeng, J., Shirihai, O. S. & Grinstaff, M. W. Degradable Nanoparticles Restore Lysosomal pH and Autophagic Flux in Lipotoxic Pancreatic Beta Cells. *Adv. Healthc. Mater.* 1801511 (2019) doi:10.1002/adhm.201801511.
2. Trudeau, K. M. *et al.* Lysosome acidification by photoactivated nanoparticles restores autophagy under lipotoxicity. *J. Cell Biol.* **214**, 25–34 (2016).
3. Arotcarena, M.-L. *et al.* Acidic nanoparticles protect against α -synuclein-induced neurodegeneration through the restoration of lysosomal function. *Aging Cell* **21**, e13584 (2022).

REVIEWER COMMENTS

Reviewer #1 (Remarks to the Author):

The manuscript has improved substantially by addressing many of the concerns raised by the reviewer and other reviewers. However, there are still some unclear parts. For instance, it is not clear what p-IR means in Fig. S2H-I, as insulin receptor phosphorylation is not mentioned elsewhere in the manuscript. It would be helpful for the authors to specify whether p-IR refers to insulin receptors or IRS proteins, and to disclose which specific site (tyrosine or serine/threonine) is being monitored. This information is crucial, as different sites of phosphorylation are differently regulated and have different functions.

The manuscript does not provide sufficient information on the specific phospho-antibodies used in the study. It would be helpful for the authors to specify the exact sites of phosphorylation being monitored for all antibodies, including p-AKT, p-GSK3b, and p-IRS (or p-IR, which is unclear). Providing this information would allow for a more accurate interpretation of the results and enhance the overall quality of the manuscript.

Finally, it would be helpful for the authors to include the catalog numbers of all critical key resources (including phospho antibodies) in the materials and methods section. Currently, the information provided in this section is incomplete, which may hinder the reader's ability to fully understand the methods used in the study.

Reviewer #3 (Remarks to the Author):

Authors have addressed my concerns adequately. I am satisfied with the revision.

REVIEWER COMMENTS

Reviewer #1 (Remarks to the Author):

Comment 1: The manuscript has improved substantially by addressing many of the concerns raised by the reviewer and other reviewers. However, there are still some unclear parts. For instance, it is not clear what p-IR means in Fig. S2H-I, as insulin receptor phosphorylation is not mentioned elsewhere in the manuscript. It would be helpful for the authors to specify whether p-IR refers to insulin receptors or IRS proteins, and to disclose which specific site (tyrosine or serine/threonine) is being monitored. This information is crucial, as different sites of phosphorylation are differently regulated and have different functions.

Response 1: Thank you for pointing this out. We have now made changes to the manuscript which specified that p-IR refers to the phosphorylated insulin receptor β , encoded by *INSR* gene. The specific phosphorylation sites being monitored are tyrosine 1162/1163. We have made changes to manuscript page 6. We have also indicated the specific phosphorylation sites in Fig. S2H and in the caption of Fig. S2.

Comment 2: The manuscript does not provide sufficient information on the specific phospho-antibodies used in the study. It would be helpful for the authors to specify the exact sites of phosphorylation being monitored for all antibodies, including p-AKT, p-GSK3b, and p-IRS (or p-IR, which is unclear). Providing this information would allow for a more accurate interpretation of the results and enhance the overall quality of the manuscript.

Response 2: We thank the reviewer for comment. We have now indicated the sites of phosphorylation being monitored for all antibodies, including p-IR (Tyr 1162/1163), p-AKT (Thr 308) and p-GSK3 β (Ser 9), in manuscript page 6. We have also indicated the specific phosphorylation sites in Fig. S2H and in the caption of Fig. S2.

Comment 3: Finally, it would be helpful for the authors to include the catalog numbers of all critical key resources (including phospho antibodies) in the materials and methods section. Currently, the information provided in this section is incomplete, which may hinder the reader's ability to fully understand the methods used in the study.

Response 3: We thank the reviewer for pointing this out. We have now provided catalog numbers for all critical key resources, including phospho antibodies, used in the materials and methods section (manuscript pages 16 - 25). Additionally, we have added a materials catalog document to the SI,

Reviewer #3 (Remarks to the Author):

Comment: Authors have addressed my concerns adequately. I am satisfied with the revision.

Response: Thank you for your comment and the prior critiques which improved the manuscript.

REVIEWERS' COMMENTS

Reviewer #1 (Remarks to the Author):

The concerns were properly addressed.